
# Assessment and application of melting layer simulations for spaceborne radars within the RTTOV-SCATT v13.1 model

Rohit Mangla[1,4], Mary Borderies[1], Philippe Chambon[1], Alan Geer[2], and James Hocking[3]

[1]CNRM, Université de Toulouse, Météo-France, CNRS, Toulouse, France
[2]ECMWF, Shinfield Park, Reading, RG2 9AX, UK
[3]Met Office, Fitzroy Rd., Exeter, UK
[4]Now at Centre for Remote Imaging, Sensing and Processing, National University of Singapore

**Correspondence:** Mary Borderies (mary.borderies@meteo.fr)

**Abstract.** Because of their high sensitivity to hydrometeors and their high vertical resolutions, space-borne radar observations are emerging as an undeniable asset for Numerical Weather Prediction (NWP) applications. The EUMETSAT (European Organisation for the Exploitation of Meteorological Satellites) NWP SAF (Satellite Application Facility) released an active sensor module within version 13 of the RTTOV (Radiative Transfer for TOVS) software with the goal of simulating both

active and passive microwave instruments within a single framework using the same radiative transfer assumptions. This study provides an in-depth description of the radar simulator available within this software. In addition, this study proposes a revised version of the existing melting layer parametrization scheme of Bauer (2001) within the RTTOV-SCATT v13.1 model to provide a better fit to observations below the freezing level. Simulations are performed with and without melting layer schemes for the Dual precipitation radar (DPR) instrument onboard GPM using the ARPEGE (Action de Recherche Petite Echelle

Grande Echelle) global NWP model running operationally at Météo-France for two different one-month periods (June, 2020 and January, 2021). Results for a case study over the Atlantic ocean show that the revised melting scheme produces more realistic simulations as compared to the default scheme both at Ku (13.5 GHz) and Ka (35.5 GHz) frequencies and these simulations are much closer to observations. A statistical assessment using more samples show significant improvement of the first-guess departure statistics with the revised scheme compared to the existing melting scheme. As a step further, this study

showcases the use of melting layer simulations for the classification of precipitation (stratiform, convective and transition) using the Dual Frequency Ratio algorithm (DFR). The classification results also reveal a significant overestimation of the rain reflectivities in all hemispheres, which can either be due to a tendency of the ARPEGE model to produce a too large amount of convective precipitation, or to a mis-representation of the convective precipitation fraction within the forward operator.

## 1 Introduction

Spaceborne radars at several frequencies (Ku:13.5 GHz; Ka: 35.5 GHz; W: 94 GHz) offer the unique capability to observe clouds and precipitation in three dimensions and at the global scale (Battaglia et al., 2020). Observed radar reflectivity (Z) profiles have been widely used by both the climate and the Numerical Weather Prediction (NWP) community, mainly for validating global NWP model outputs (Bodas-Salcedo et al., 2008, 2009, 2011), and a few studies also investigated the assimilation





of these data (Fielding and Janisková, 2020; Ikuta et al., 2021; Kotsuki et al., 2023). In particular, the Cloudsat cloud profiling radar (CPR) (Stephens et al., 2002) has been used to validate global model forecast quality in previous works (e.g. UK Met Office Unified Model (MetUM) by Bodas-Salcedo et al. (2008); Global Climate Model (GCM) by Marchand et al. (2009); and European Centre for Medium-range Weather Forecast (ECMWF) Integrated Forecasting System (IFS) by Di Michele et al. (2012)). Similarly, the Precipitation radar (PR) onboard the Tropical rainfall measuring mission (TRMM) has been used to evaluate precipitating cloud types and microphysics in a cloud resolving model (CRM) in Matsui et al. (2009); Li et al. (2008). Finally, the Dual-frequency precipitation radar (DPR) onboard the Global precipitation measurement (GPM) mission (Hou et al., 2014) has been used as reference by Mai et al. (2023) to evaluate cloud microphysical properties in the Weather research forecasting (WRF) model. A review of different operational applications of spaceborne precipitation radars is available in the International Precipitation Working Group (IPWG) report (Aonashi et al., 2021).

Using observations to evaluate a model directly requires the use of a radar simulator to transform model variables into radar reflectivities at model levels, the so-called model-to-satellite or forward approach, which is commonly used in data assimilation applications and climate model evaluation studies (Ringer et al., 2003). Many such simulators exist for example Quickbeam (Haynes et al., 2007), Joint-Simulator (Hashino et al., 2013), ZmVar (Fielding and Janisková, 2020), Passive and Active Microwave TRAnsfer (PAMTRA) (Mech et al., 2020), Community Radiative Transfer Model (CRTM) (Weng et al., 2005) and Goddard Satellite Data Simulator Unit (G-SDSU) (Matsui et al., 2013, 2009). However, the Radiative Transfer for the television infrared observation vertical sounder (RTTOV) (Saunders et al., 2018) model, which is widely used in model-to-satellite simulations of passive radiances, particularly for NWP, has not provided a radar capability until very recently. This study presents the new radar simulator available in the version 13 of RTTOV-SCATT which benefits from state-of-the art developments for passive microwave simulations within clouds and precipitation (Geer et al., 2021) .

Radar signals from precipitation radars are very difficult to model in the mixed phase region (also called melting layer), where frozen hydrometeors are in transition from frozen to raindrops (Zhang et al., 2008). In the melting layer, the maximum size snowflakes are first transform into wet flakes, then to raindrops of smaller sizes of equivalent mass and less number density as compared to original flakes (Galligani et al., 2013). There exist large differences in permittivity between the two phases, which cause the sudden enhancement of reflectivity known as the 'bright band' (hereafter, BB), most prominently in stratiform precipitation (Giangrande et al., 2008; Klaassen, 1988; Karrer et al., 2022). In addition to accurately simulate the bright-band at the freezing level to provide realistic comparisons between model and simulations, it is also important for deriving realistic simulated profiles of attenuated reflectivity. Indeed, a too strong bright band attenuates the radar signal and impacts the attenuated reflectivity in the rain layer below, which could result in an underestimation of the rainfall.

In the past years, the simulation of the melting layer has been extensively investigated for ground based radars using two dimensional physical models (Szyrmer and Zawadzki, 1999), but rather limited for spaceborne radars (Augros et al., 2016; Klaassen, 1988; Boodoo et al., 2010; Iguchi et al., 2014; Das et al., 2023). For instance, Kollias and Albrecht (2005) simulated melting layer reflectivities for Cloudsat CPR, and observed a small *decrease* in reflectivity (1-2 dB) just above the freezing level due to Mie backscattering effects, which they refer to as the *'dark band'*. Similar results were also found in the conclusions of Sassen et al. (2007, 2005); Lhermitte (1988). Olson et al. (2001) used a steady state melting layer model to simulate TRMM



PR reflectivities and noticed that melting precipitation can result in a 5 dB increase in the layer-mean reflectivity at Ku-band.
Bauer (2001) presented a melting layer scheme to compute scattering properties for both passive and active instruments, which
is already available in RTTOV-SCATT. This scheme treated hydrometeors as homogeneous spheres. Geer and Baordo (2014)
already introduced non-spherical shapes for frozen hydrometeors in RTTOV-SCATT for providing accurate simulations in
the ice region, but retains a Mie sphere representation of the melting layer. A few studies (Johnson et al., 2016; Petty and
Huang, 2010; Botta et al., 2010) reported that non-spherical particle shapes better represent the early stage of melting where
hydrometeor properties change very rapidly, but this is not considered in this study.

It is well known from past studies (Kollias and Albrecht, 2005; Romatschke, 2021) that while W-band radars are excellent
for observing clouds and light precipitation, they are not best-suited for observing heavy precipitation due to greater attenuation
and multiple scattering effects compared to lower frequency radars. However, the signal attenuation at Ka band is smaller and
is almost negligible at Ku band in the melting layer, which allows the bright-band to be seen at Ku and Ka bands(Galligani
et al., 2013; Zhu et al., 2022). Therefore, this study focuses on Ku and Ka bands to simulate bright band signatures, and to
assess the benefits obtained by the use of a revisited parametrization of Bauer (2001).

In the observation world, the bright band detection has several applications. One of them is precipitation type classification
(Iguchi et al., 2010). Le and Chandrasekar (2012) proposed a Dual frequency ratio (DFR, e.g. difference between Ku and
Ka band reflectivities) classification algorithm which is based on the vertical variation of the DFR within and below the
melting layer. If the melting layer is detected, then the DFR algorithm is able to classify precipitation into three categories (i.e.
Stratiform, Convective and Transition). In the areas where both frequencies are not available, the DFR algorithm is associated
with another single frequency algorithm for the precipitation type classification within the GPM DPR level 2 product (Huffman
et al., 2018; Iguchi et al., 2010; Kobayashi et al., 2021; Awaka et al., 2016). The vertical variation of DFR is a base for the
development of many other algorithms for DPR observations like Graupel and Hail detection algorithm by Le and Chandrasekar
(2021), and surface snowfall detection algorithm by Le et al. (2017). Motivated by the success of the DFR algorithm on
real observations, this study investigates whether the bright band parametrization of RTTOV-SCATT has reached a sufficient
degree of realism that the DFR algorithm can also be applied to NWP model profiles and further, whether it can support the
precipitation classification of NWP model profiles with a reasonable degree of accuracy.

Therefore, the present study has two objectives. The first objective is to present the upgrade of the melting layer parametrization scheme in the RTTOV-SCATT model v13.1, with the upgrade now available in v13.2. This objective also validates the
simulations for the DPR Ku-Ka instrument by comparing observations to simulations. The second objective is to investigate
whether if the bright band simulations for Ku and Ka band are realistic enough that they can be used for applications like
precipitation type classification. The paper also functions as a first complete documentation of the RTTOV radar simulator in
the peer-reviewed literature. The paper is organized as follows: section 2 briefly describes the two datasets used in the study
from the GPM/DPR instruments as well as the ARPEGE NWP model, section 3 describes the RTTOV-SCATT radar forward
operator and melting layer parametrization schemes within the forward operator, section 4 presents the impact of melting layer
on reflectivity simulations on a case study and with colocated samples between the GPM/DPR and ARPEGE over a 2-month





period. The DFR algorithm for precipitation classification is explained and applied to ARPEGE in section 5 followed by the conclusions and perspectives in section 6.

## 2 Data and Models

### 2.1 GPM/DPR observations

The GPM mission was launched on 28th February 2014. It carries a dual-band (Ku and Ka) precipitation radar (DPR) that provides three-dimensional observations of clouds and precipitation to a wide range of latitudes (67°S-67°N) (Hou et al., 2014; Skofronick-Jackson et al., 2017). Both radars have a center beamwidth of 0.71° and footprint resolution of 5 km. The swath width of the Ku band radar is of 245 km and the one of the Ka band radar is 120 km. The minimum detectable signals are of 18 dBZ and 12 dBZ for Ku and Ka bands respectively (Hamada and Takayabu, 2016).

Version 6 of the 2A DPR product, which was created by combining Ku and Ka frequency radar echoes within the same scattering volume, is used in this study. The 2A DPR products have been used in many past studies for calibration, validation and classification purposes (Kotsuki et al., 2014; Zhang and Fu, 2018; Sun et al., 2020). For example, Sun et al. (2020) used 2A DPR product to classify stratiform and convective pixels to study vertical structures of typical Meiyu precipitation event over China. The inner swath pixels (25) in match swath (MS) mode are used in this study, where both Ku and Ka band are on the same grid and viewing the exactly same area and having the same vertical resolution of 250 m. Further details about different modes of DPR scan pattern are given in Hou et al. (2014). It has to be noted that MS pixels are oversampled to 125 m vertical resolution in the 2A DPR product. For this study, attenuated Ku and Ka reflectivity profiles, precipitation type, and quality flags are used.

Figure 1 shows an example of a single orbital file on 2nd January, 2021. For this particular orbit, the sensor followed the path starting at the black point and ending at the gray point. The black box depicted the Atlantic ocean is chosen for a case study as a first step before running a statistical study (more details are given in subsection 4.2).





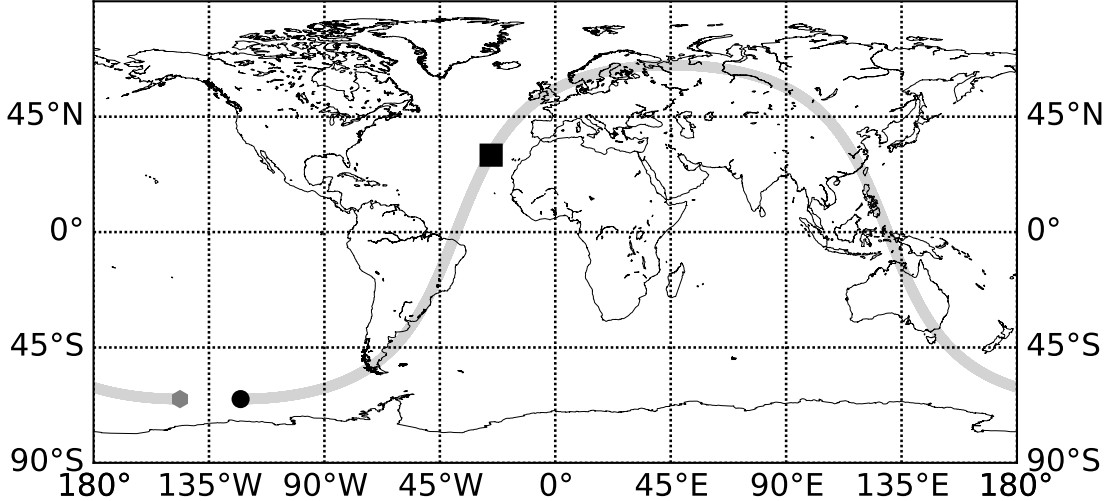

**Figure 1.** Ground track of one orbit of GPM for 2nd January, 2021. The black dot corresponds to the beginning of the orbit file and the gray dot to the end of it. The box represents the location of a single cloud over the Atlantic ocean, which is used in subsection 4.2 for the case study

## 2.2 ARPEGE Global NWP Model

This study uses the ARPEGE NWP global model, which is operational at Météo-France since 1992 (Courtier, 1991). The latest operational configuration is used which is characterized by a spectral resolution of T1798, used with a stretched and tilted grid giving a horizontal resolution of 5 km across Europe to 25 km at the antipodes (New Zealand). Further details on the ARPEGE global NWP model are given in Bouyssel et al. (2022), and only briefly mentioned here. The model is characterized by 105 vertical levels, from ∼10m above the ground to model top at 0.1 hPa (∼70 km). The model uses a 6h cycling 4D-Var

incremental data assimilation scheme to generate analyses four times per day. The Ensemble of Data Assimilation of ARPEGE (AEARP, Desroziers et al., 1999) is used to compute the background error covariance matrix. ARPEGE makes use of 6 hydrometeor types to represent clouds and precipitation, 4 of them are prognostic variables (stratiform snow, stratiform rain, cloud liquid water and cloud ice) and their evolution is governed by the Lopez (2002) microphysical scheme and two of them (convective snow and convective rain) are diagnosed variables and governed by the deep convective scheme of Tiedtke (1989).

In this study, the ARPEGE model is setup for two different months (1) June 1- 30, 2020 [summer in Northern Hemisphere (NH) and winter in Southern Hemisphere (SH)], (2) January 1-31, 2021, [winter in NH and summer in SH].

## 2.3 Collocation between ARPEGE model and observations

The choice is made in this study to compare observations and model forecasts in the short range only, at +6h forecast range. Simulations and observations are vertically, horizontally and temporally collocated. To avoid as much as possible time mis-

matches between observations and simulations, only the observations which are available within ±30 minutes surrounding the





validity time of the +6h forecasts are used for the comparisons against simulated observations. Spatially, the nearest ARPEGE latitude-longitude grid point is allocated to each observed location. Di Michele et al. (2012) performed the averaging of radar profiles corresponding to the model grid box. However, averaging introduces additional uncertainty, especially when the clouds are non-homogeneous within a grid point. Performing such an averaging is also difficult for ARPEGE as the horizontal resolu-
tion varies with the latitude and longitude. The choice was made to not average DPR data at the model resolution in the study but to keep this information in mind when interpreting the results by hemisphere and geographical zones.

The reflectivity simulations with the ARPEGE model are generated at 105 height levels. However, DPR observations are available at 176 height levels (125 m each) at a finer vertical resolution. To remove the discrepancies between observed and model height levels, simulations are interpolated linearly in geometric height and in reflectivity ($mm^6.m^{-3}$).

## 140  3   RTTOV-SCATT V13.1 forward operator

This study uses the RTTOV-SCATT (v13.1) radiative transfer model for the simulation of GPM/DPR reflectivities both at Ku and Ka band (Geer et al., 2021). Note that RTTOV-SCATT also include the tangent-linear (TL) and adjoint (AD) versions of the radar simulator. However, as this study is focused on the forward operator, they are not described here.

The inputs of the forward operator are profiles of temperature, pressure, specific humidity, hydrometeor contents, hydrom-
eteor fractions and sensor parameters (e.g. radar frequency, grid location, and viewing angle). For this study, a new feature of RTTOV allowing to set a specific number of hydrometeors has been used: the configuration used therefore takes into account the six hydrometeors classes (described in subsection 2.2) of ARPEGE, instead of only five hydrometeors in the default configuration. The forward operator uses the radar equation (Equation 1) for the simulation of attenuated and unattenuated reflectivities at each radar gate. Assuming the six categories of hydrometeors (i) and their particle size distributions ($N_i$), the
reflectivity ($mm^6 m^{-3}$) is computed by integrating the backscattering cross-section $\sigma_i(D)(m^2)$ over the Particle Size Distribution (PSD) within the size limits $D_{min}$ to $D_{max}$. The total reflectivity is obtained after summing up the contributions from all six hydrometeors at each range gate which is at a distance $R\ (m)$ from the radar. The signal value is very large in number, therefore converting into logarithmic scale (dBZ) is a standard way of using radar products in meteorology.

$$Z(R)[dBZ] = 10\log_{10}\left(10^{18}\frac{\lambda^4}{\pi^5 \mid K_w \mid^2}A(R) \times \sum_{i=1}^{6} f_i(R)\int_{D_{min}}^{D_{max}} \sigma_i(D)N_i\{D, W_i(R)\}\,dD\right) \tag{1}$$

Here, $\lambda$ (m) is the radar wavelength (Ku $\approx$ 0.022 m, and Ka $\approx$ 0.0086 m), $\mid K_w \mid^2$ is a factor depending on the permittivity of liquid water, $W_i(R)\ (kg/m^3)$ is the in-cloud water content of the hydrometeor $i$, $f_i(R)$ is the fraction of the grid box occupied by each hydrometeor (see Equation 4) and $A(R)$ is the two way attenuation which is computed according to Equation 3. Further details on the radar equation are available in Geer et al. (2021); Borderies et al. (2018); Johnson et al. (2016). The in-cloud water content is related to the grid box average water content $W_{av,i}(R)$ by the hydrometeor fraction:

$W_i(R) = W_{av,i}(R)/f_i(R).$ \hfill (2)





The attenuation is computed along the downward and backscattered upward radar beam as follows:

$$A(R) = \exp\left(-2\int\limits_0^R \sum_{i=1}^6 f_i(R) \int\limits_{D_{min}}^{D_{max}} C_i(D)N_i\{D,W_i(R)\}\,dD + a(R)dR\right) \quad (3)$$

Here, $C_i(D)$ is the extinction coefficient and 2 is the multiplying factor to account for the 2-way path attenuation. Attenuation by moist air and hydrometeor is accounted in *A(R)* computation, where $a(R)$ represents the gas absorption coefficient. The
hydrometeor fraction in the attenuation represents a 'single column' model in the terminology of Fielding and Janisková (2020) and a future update could be to use the double column approach as proposed in that work. The RTTOV-SCATT model does not account for multiple scattering effects but will be considered in the future. This physical process could be important at W band (Battaglia et al., 2011) and Ka band (Battaglia et al., 2015) in heavy precipitation events.

To simulate the reflectivity, bulk scattering properties (backscattering and extinction coefficients integrated over the PSD)
are stored in a look-up table as a function of temperature, frequency and water content for each hydrometeor type (Bauer, 2001; Geer and Baordo, 2014; Geer et al., 2021). These lookup tables are generated by specifying particle shape and its associated mass-diameter relationship, PSD, and (for spheres) permittivity model for each type of hydrometeor. The operator offers the Liu (Liu, 2008) and ARTS (Eriksson et al., 2018) databases of non-spherical shapes to represent realistic frozen hydrometeor habits, which were generated using the Discrete Dipole Approximation (DDA). For this study, the default DDA based shapes
(see Table 1) are used for snow and graupel hydrometeors as suggested by the studies of Geer (2021) and Geer et al. (2021). Similar DDA based shapes have already been used for simulating the GPM/DPR instrument in the past studies (Liao et al., 2013; Johnson et al., 2016). Spherical shapes are assumed for melted particles, including fully liquid hydrometeors (raindrops and cloud liquid water) as well as the melting layer. Mie theory is used to compute the optical properties in these cases. The densities for frozen hydrometeors are computed using the assumed mass-density relationship; for liquid hydrometeors the
density is assumed equal to 1000 $kg/m^3$. The PSDs for snow and graupel hydrometeors are represented by Field et al. (2007), and a Modified Gamma Distribution (hereafter, MGD) for ice particles. The Marshall and Palmer (1948) (hereafter, MP) is used for convective and large-scale rain, and a MGD is used for water cloud. It is noted that a MGD PSD is a function of 4 parameters (number density $(N_0), \mu, \Lambda, \gamma$) which are different for each particle type. For cloud ice, $\mu = 2, \Lambda = 2.13*10^5$ is chosen, whereas $\mu = 0, \Lambda = 1*10^4$ is assigned to cloud water. The $N_0$ is free parameter for both particles. These parameter
values are taken from Geer et al. (2021). The permittivity is required for spherical particle models including the current melting layer formulation and is computed following this model: (1) Maxwell-Garnett model extended for ellipsoid inclusions for snow and graupel, and Mätzler (2006) model for ice hydrometeors down to the top of melting layer (Garnett, 1904; Meneghini and Liao, 2000; Bohren and Battan, 1982), (2) within the melting layer, if active, the Fabry and Szyrmer (1999) model number 5 (detail described in subsection 3.1), and (3) the Rosenkranz (2015) model is used for liquid water drops.

For each hydrometeor type $i$, the hydrometeor fraction profile $f_i(R)$ needs to be provided to Eqs. 1, 2 and 3. In this study, the fraction used for stratiform hydrometeors $pfrac$ is derived from the cloud cover profile from the NWP model using the same formula as in the ECMWF IFS model (Park, 2018), except that the evaporation is not accounted for here. At a given



level [j+1], *pfrac* is calculated according to Equation 4. It is a function of the cloud cover $CC$ at the current level [j+1] and the previous level [j] located above it, and the *pfrac* at $j^{th}$ level.

$$pfrac[j+1] = 1 - \frac{(1 - pfrac[j])(1 - max(CC[j], CC[j+1]))}{1 - min(CC[j], 1 - e^{-6})} \qquad (4)$$

Here, $CC$ is the cloud cover profile and $j$ is the height level.

The *pfrac* profiles computed using Equation 4 are assigned as the hydrometeor fractions $f_i(R)$ for for stratiform snow and rain. $f_i(R) = 5\%$ is assigned to convective hydrometeor fractions for the entire profile. This study chooses 5% because this value is being used operationally at ECMWF and Météo-France for passive microwave observations. The $CC$ profiles are used for cloud liquid water and cloud ice water fractions (see Table 1).

For a given ARPEGE grid cell, the bulk (PSD integrated) backscattering and extinction coefficients are interpolated from the lookup tables given the temperature and in-cloud hydrometeor water content $W_i(R)$. Then, bulk backscattering and extinction coefficients are used to compute the attenuated (AZef; Eq. 1) and unattenuated (Zef; Eq. 1 with $A(R) = 1$) reflectivities at both Ku and Ka bands.

**Table 1.** Default configuration for the 6 hydrometeors settings used in this study. Note that this table applies to non-melting hydrometeors only. Melting hydrometeor properties are described in subsection 3.1.

| Hydrometeor type | PSD | Shape | Hydrofraction | Permittivity |
|---|---|---|---|---|
| LS Rain | MP | Spherical | Diagnosed from $CC_S$ | Rosenkranz (2015) |
| Convective Rain | MP | Spherical | 5% | Rosenkranz (2015) |
| Cloud liquid water | MGD | Spherical | cloud cover ($CC$) | Rosenkranz (2015) |
| Cloud ice | MGD | Large column aggregates | cc | Mätzler (2006) |
| Snow | F07T | Large plate aggregates | Diagnosed from $CC_S$ | Maxwell-Garnet Model |
| Graupel | F07T | Column | 5% | Maxwell-Garnet Model |

## 3.1 Melting layer parametrization of Bauer (2001)

The bulk backscattering and extinction coefficients in the melting layer depend on various factors such as permittivity, PSD, shape, phase, etc, which can significantly alter the reflectivity computation. One of the major contributor is the permittivity which can induce significant changes in the scattering coefficients (Fabry and Szyrmer, 1999). When using a spherical model for the melting hydrometeors as done here, the permittivity and the diameter of the sphere are the main controls over the simulated scattering properties. Therefore a major focus of this study is on an accurate representation of the permittivity within the melting layer.

The RTTOV-SCATT model includes the possibility of activating the Bauer (2001) melting model for snow and graupel including the two-phase coated sphere as described in Fabry and Szyrmer (1999, their model 5). In this original parametrization,





the melting layer width is set to 1000 m, with 100 height levels ranging from the freezing level $T_{fl}$ ($T_{fl}$=273 K) to the bottom

of melting layer defined by its temperature $T_{ml}$. The melting process, including the melting fraction $f_m$, is computed based on the model of Mitra et al. (1990).

The model starts from the relevant PSD for the completely frozen particle (eg. Field et al. (2007)) and computes the properties of these particles as they fall and melt, always using spherical particle assumptions. It should be noted that the PSDs used for snow and graupel at the top of the melting layer in version 13 of RTTOV-SCATT (eg. Field et al. (2007) PSD) are not consistent

with the ones used in the original formulation of Bauer (2001) (eg. Marshall-Palmer PSD). However, Geer et al. (2021) suggests that the differences between the two PSDs are not that large, and that the Field et al. (2007) may even have reduced the number of larger particles. Another difference with the parametrization of Bauer (2001), is that density and mass of the frozen particles is now also specified by the mass and density distributions implied by the frozen particle shape choice (see Table 1), rather than the mass-density relationship used by Bauer (2001)

At the top of the melting layer, (i.e. $T_{fl}$), properties of frozen hydrometeors are represented by ice in a matrix of air [[ice], air]. As the melting increases, the properties change according the percentage of melted hydrometeor $f_m$. In the two-phase model, the hydrometeors are represented by two phases in a coated sphere model comprising an inner core and outer coat. The core is modelled as air inclusions in a matrix composed by a mixture of ice inclusions in water matrix [air,[[ice], water]]. The coat is composed of ice inclusions in a water matrix in a air matrix [[[ice], water], air]. The percentage of core and coat

varies with the melting fraction $f_m$. The higher is $f_m$, the higher is the coat percentage. The same melting model has also used for the simulation of W-band Cloudsat radar reflectivity in Kollias and Albrecht (2005). Once the permittivity of the melted hydrometeor has been assigned to their respective height, the Mie scattering calculations are performed to compute the scattering properties (backscattering and extinction). They are then integrated and averaged over the full temperature ranges (from 273K to 275K). Details are given in Appendix A.

In Bauer (2001) and up to version 13.1 of RTTOV-SCATT, if the melting layer was active, the optical properties were stored in the 273 K temperature bin of the lookup tables for graupel and snow hydrometor types, whereas the lower temperature bins (272 K and below) provided optical properties based on the standard frozen hydrometeor microphysical choices. However, in most NWP models there are also frozen hydrometeors at positive temperatures. An example within the ARPEGE global model is given in Figure 2, which shows a significant occurrence of graupel at warm temperature until 277 K. Therefore,

subsection 3.2 proposes modifications which allow to have a smoother and more accurate vertical representation of the melting process, in compliance with the existence of melting particles at warm temperatures.





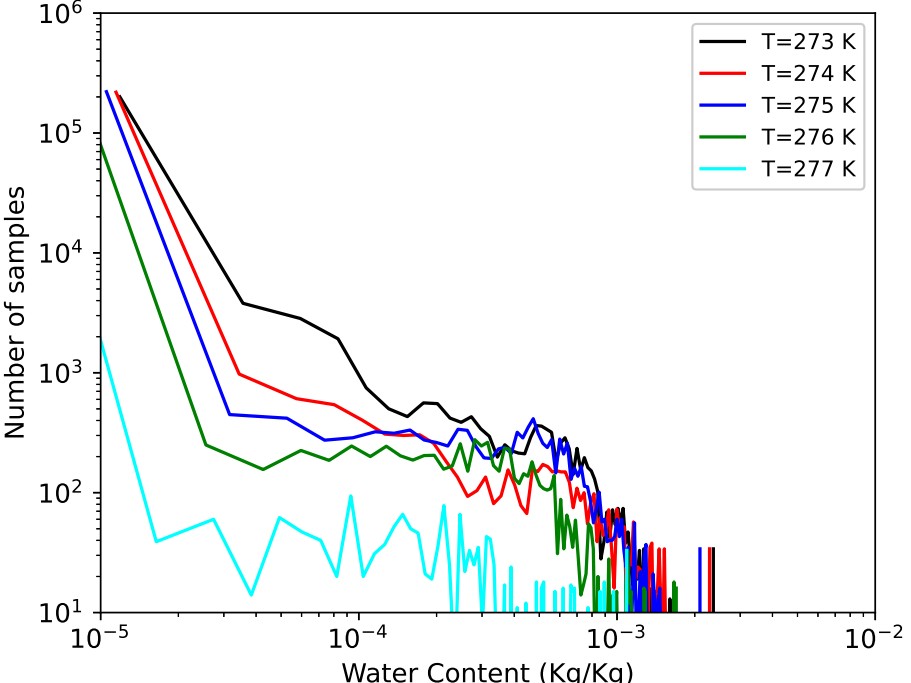

**Figure 2.** Distribution of graupel content at different warm temperatures from the ARPEGE global NWP model. The ARPEGE samples used here are collocated on a single GPM orbital file on 2nd January, 2021 (shown in Figure 1). It is noticed that significant graupel content exists at warm temperature until 277 K and should be accounted for in melting layer simulations

### 3.2 Revised version of Bauer, (2001) parametrization

First, in the revised version of the melting layer, the temperature at the bottom of the melting layer $T_{ml}$ is chosen in accordance to the graupel content distribution shown in Figure 2. Indeed, it has been observed that graupel content is significantly present until 277 K and assumed that graupel is fully melted after 277 K and converted into raindrops. Therefore, $T_{ml}$ is now set to 277 K for the rest of this study. Secondly, the revised melting layer is parametrized across five sub-melting layers from 273 K to 277 K in steps of 1 K, which allows to interpolate the scattering coefficients in the lookup tables consistently with the temperature provided by the NWP model. Finally, new lookup tables of scattering coefficients including positive temperature bins for melting hydrometeors are computed in accordance with the physical processes involved in the melting layer. Further details are provided in Appendix A.

As will be seen in the following sections, the new melting layer provides a less intense but broader melting layer that is better agreement with observations. As shown in Appendix A, this improved agreement comes partly through a reduction in backscatter that has been generated by an ad-hoc scaling of the optical properties introduced by splitting the layer into five parts. This ad-hoc scaling was originally unintentional, but it generated such good improvements in the melting layer representation that it was adopted anyway.





# 4 Impact of melting layer on reflectivity simulations

## 4.1 Impact of melting layer on the lookup tables

Figure 3 compares the Ku (top panels) and Ka (bottom panels) band reflectivities, derived from the backscattering coefficients of the lookup tables, with three different scenarios: (1) No bright band (hereafter, $NO_{BB}$), (2) bright band with the original

Bauer melting scheme (hereafter, $def_{BB}$), and (3) bright band with revised Bauer scheme (hereafter, $rev_{BB}$). Figure 3a and c show the comparison of radar reflectivity as a function of snow and graupel water content between $NO_{BB}$ and $def_{BB}$ at 273 K. As expected, one can see in Figure 3a and c that the snow and graupel reflectivities increase by 15 dB and 10 dB at Ku and Ka band respectively with the activation of the melting layer (see the differences between different color lines).

    Figure 3b and d compare the graupel reflectivities with the $def_{BB}$ and $rev_{BB}$ scenarios in both bands. It is observed that

the reflectivity with the $rev_{BB}$ scheme is significantly reduced at 273 K, confirming that the $rev_{BB}$ approach allows to split the melting processes across different temperature bins in the melting layer, which are shown with the different color lines.

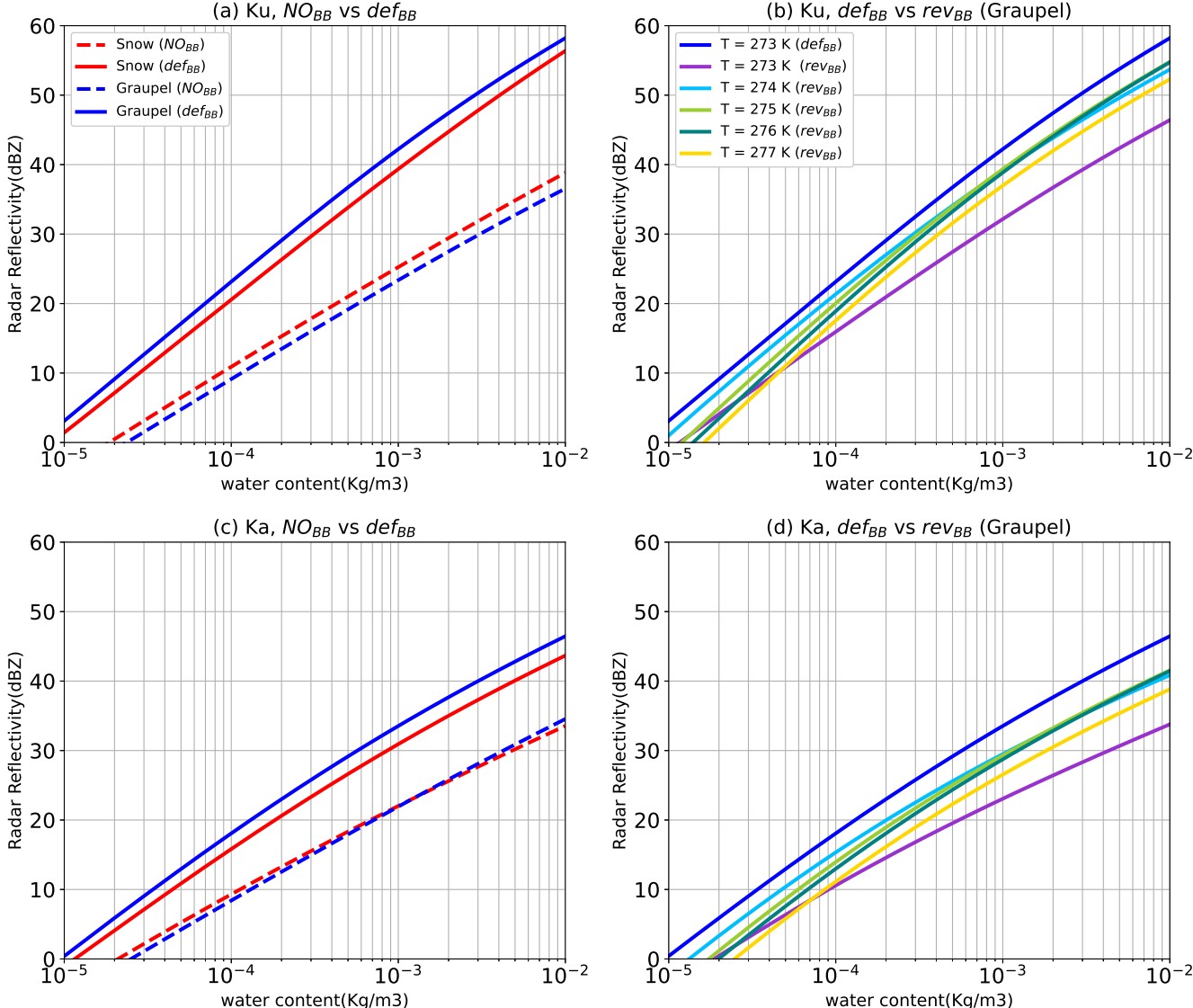

**Figure 3.** Radar reflectivity as a function of water content at Ku and Ka band. (a, c) Comparison of snow and graupel reflectivities without melting layer ($NO_{BB}$) and with default Bauer parametrization ($def_{BB}$) at T=273 K. The reflectivity increases by 15 dBZ and 10 dBZ in Ku and Ka band respectively after the activation of melting layer. (b, d) Graupel reflectivities are compared with $def_{BB}$ scheme (blue color) and the revisited version of Bauer parametrization ($rev_{BB}$) at extended warm bins.

Similarly, extinction coefficients for snow and graupel also increase by a factor of 100 both for Ku and Ka bands as shown in Figure 4, and the magnitude is larger at Ka band. This is expected because Ka band radars are more subject to attenuation by precipitation particles than Ku-band radars.

**Figure 4.** Similar to Figure 3, Extinction coefficients ($m^{-1}$) as a function of the content are compared with the three scenarios.

## 4.2   Case Study

The RTTOV simulations are first performed using the default configuration (described in Table 1) for a case study mentioned in Figure 1 and with the three scenarios (i.e $NO_{BB}$, $def_{BB}$, and $rev_{BB}$). For the case study, the cloud profiles were selected in the area 28°N-32°N; 28°W-22°W (also shown with the gray box in Figure 1) over the Atlantic ocean. This case study was selected because, firstly, it is over ocean, which avoids the disparities associated with the relief in the rainy levels between neighbouring reflectivity profiles. It was also selected because of its observed bright-band signature at approximately 3.5 km





of altitude. Lastly, this precipitating system is well predicted in ARPEGE, making it an ideal candidate for comparing observations with simulations made with different parametrizations of the bright-band.

Figure 5 (resp. Figure 6) compares the horizontal cross-section of the selected piece of orbit at three different heights (i.e. 2,
3.25 and 6 km from left to right) at Ku (resp. Ka) band for the observation (top panels) and the simulations. Overall, the spatial structures of the simulated cloud is well matched with observations in all three scenarios. One can observe that the intensity is also of the same order of magnitude in the ice levels at $\sim$ 6 km. However, at 3.25 km in the melting layer, they differ by $\sim$ 10 dB. Comparisons of e and h panels of Figures 5 and 6 show large overestimation in Ku band reflectivities and relatively lower in Ka band with $def_{BB}$ scheme. This overestimation is slightly reduced by a factor of 2 to 4 dB when the simulations
are performed using the $rev_{BB}$ scheme. On the contrary, significant attenuation is noticed at a rain level at 2 km (relatively higher in Ka band) as a result of a strong bright band, which is further discussed by seeing the vertical structure in Figure 7.

Figure 7 shows the time-height vertical cross-section of observed and simulated reflectivities for a cross-section around the centre of the cloud (shown by dashed centre line in Figure 5 and 6 ). It should be noted that the vertical structure of the reflectivity is in reasonable agreement with the observations in both snow and rainy levels. In the melting region, simulated
bright-band reflectivities are too strong with the $def_{BB}$ scheme. With the $rev_{BB}$ scheme, this overestimation is reduced by an order of approximately 5 dB. Large differences with the observations remain, which can be explained by other assumptions in the forward operator (e.g. spherical particle shape, PSD or precipitation fraction). The extremely strong bright-band signature using $def_{BB}$ scheme causes substantial attenuation in the rainy levels, especially at Ka band (panel g). Contrarily, the $rev_{BB}$ scheme not only improves the bright band reflectivity but also reduces the attenuation affecting the reflectivities from the rainy
levels. One can notice a strong rain cloud at Ku observations (panel 7a) probably associated with mixture of stratiform and convective rain (near 30.09°N). However, this cloud has been shifted to the right side with higher intensity (near 30.48°N) in simulations (panel 7b-d). Such cloud mislocation error exists in global NWP model, because it is difficult to forecast cloud and precipitation at right location with the right intensity (Fabry and Sun, 2010). The low precipitation fraction $f_i(R)$ (5%) used for convective hydrometeors is also one factor for the overestimation. As stated in Equation 1 and Equation 2, the in-cloud
water content is normalized by $f_i(R)$, this causes the change in shape of PSD. A sensitivity study to the precipitation fraction is shown in Appendix B for the case study.

A single profile with an observed bright-band signature is also diagnosed from the experimental cloud (the dashed black line on Figure 7 corresponds to its location) and is shown in Figure 8. The observed profile is depicted in gray, and the attenuated (panels a, b, e, f) and corrected (panels c, d, g, h) simulations under three scenarios by the dotted, dash-dot and solid lines
respectively. The Ku band simulated profiles indicate that $def_{BB}$ scheme overestimates the bright band reflectivity that results into large observed minus simulated reflectivities (or first guess (FG) departure) of approximately -15 dB maxima. However, $rev_{BB}$ scheme produces a smoother transition of the reflectivities within the melting layer and is found in better agreement with the observations with smaller FG departure ($\sim$ -9 dB maxima). One can notice that both AZef and Zef simulations are overestimated in the rainy levels at Ku band. This could be due to the model (overestimation of the rain) or to a misrepre-
sentation of the rain within the forward operator (e.g. precipitation fraction and/or PSDs). Interestingly, the overestimation is





larger (panels a, b) when the bright band is not simulated. There is an artificial improvement of the bias when the bright band signal is too strong, which explains why $def_{BB}$ is better in the rainy levels. Indeed, a large bright band signal increases the extinction coefficient, which significantly increases the attenuation in the rainy levels. On the other hand, Zef at Ka band shows the opposite behavior (i.e. underestimation) in the rainy levels. This may be due to strong attenuation with rain particles at

Ka band. Furthermore, the profile of simulated attenuated reflectivity AZef indicates a very high underestimation of the rain reflectivity ($\sim$ 10 dB) by $def_{BB}$ scheme, followed by slightly reduced underestimation ($\sim$ 3 dB) using $rev_{BB}$ scheme, and almost negligible with $NO_{BB}$ in comparison to observed reflectivity. Results from a case study using a small sample size shows the positive benefits of the $rev_{BB}$ scheme in the melting levels as well as in the rainy levels at both Ku and Ka band frequencies.







**Figure 5.** Horizontal cross-section over the selected polygon at 2 km, 3.5 km and 6 km heights for Ku band. The top panel (a-c) shows the observations, (d-f) shows the simulation without bright band ($NO_{BB}$), (g-i) shows the simulated reflectivities with bright band using $def_{BB}$ scheme, while (j-l) shows the simulations with $rev_{BB}$ scheme. The centre dotted line represents the three-dimensional cross-section and the cross mark represents the location of the single profile which is used for diagnosis later in this section.





**Figure 6.** Same as Figure 5, but for the Ka band.





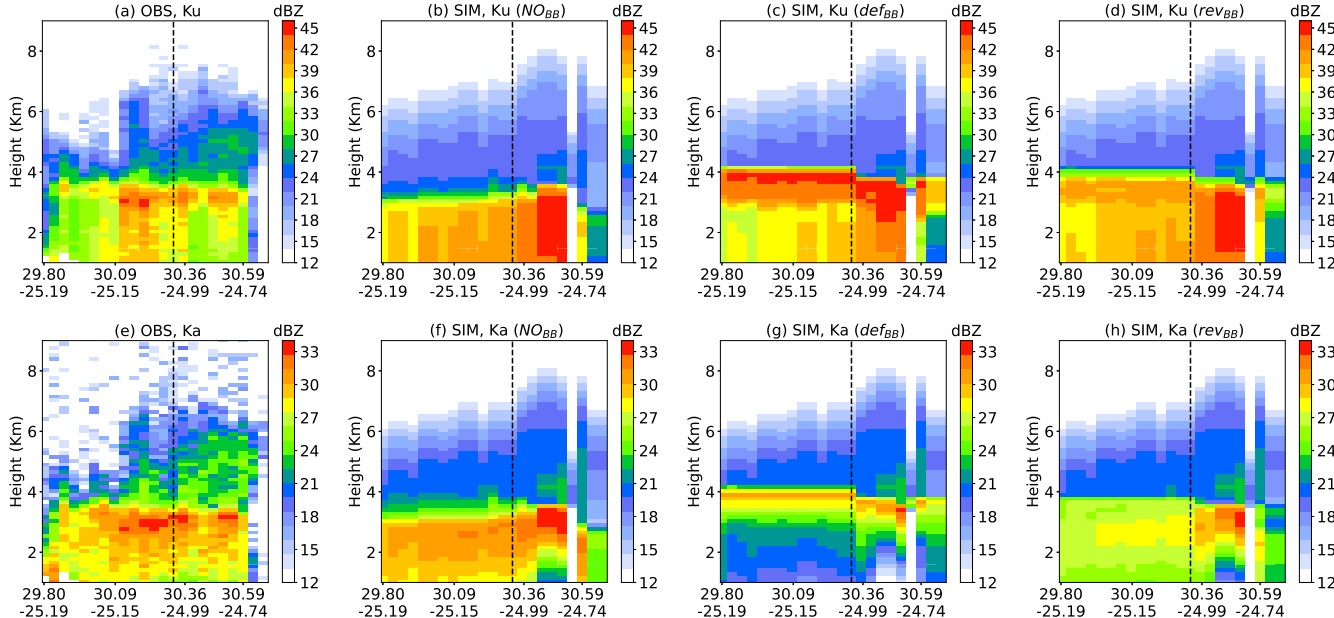

**Figure 7.** (a,e) Vertical cross section of the selected cloud in the observations and (b-d and f-h) simulations with three scenarios as mentioned above: $NO_{BB}$, $def_{BB}$, and $rev_{BB}$ configuration for Ku and Ka band respectively. The dashed black line is the single profile used for the diagnosis.

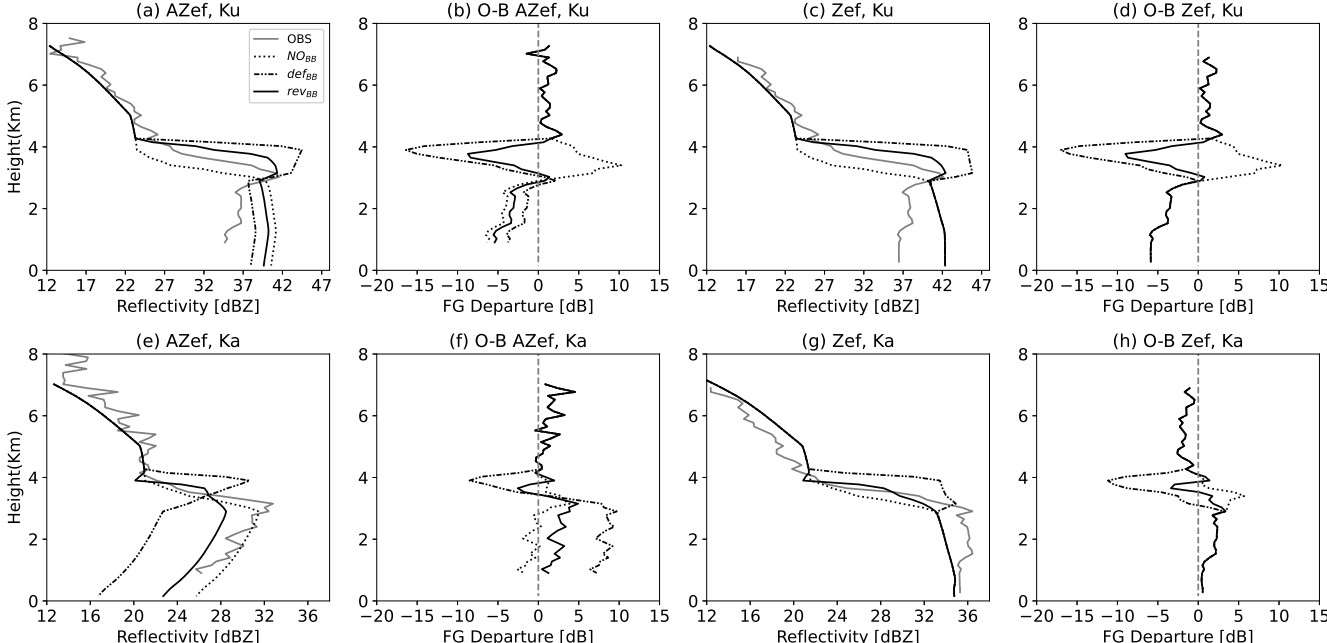

**Figure 8.** (a,e) An attenuated reflectivity (AZef) and (c,g) unattenuated reflectivity (Zef) profile is diagnosed from the experimental cloud at Ku (top panels) and Ka band (bottom panels) and compared with RTTOV simulated profile under three different scenarios. The first guess departure at both frequencies is also shown in panel (b,f) and (d,h) for AZef and Zef respectively.





## 4.3    Statistical assessment over a one-month period for two seasons


Subsection 4.2 demonstrated the positive impact of the $rev_{BB}$ on a case study. The impact is now shown over a longer period. This study performed simulations of DPR reflectivities for a one month period of two seasons i.e. June, 2020 and January, 2021 using the $NO_{BB}$, $def_{BB}$, and $rev_{BB}$ configurations. Only samples where both observed and simulated attenuated reflectivities are above the radar sensitivity of 12 dBZ are considered for computing the first guess departure statistics. Figure 9 and Figure 10

show the vertical distribution of FG departure statistics with statistical hypothesis test (Levene's and Welch's t-test). The standard deviation is shown on the left panels, bias in the right panels in the NH region (23.43°N to 60°N: top panel), tropics (-23.43°N to 23.43°N: mid panel) and SH region (-60°N to -23.43°N: bottom panel) at Ku and Ka band respectively. The Levene (Levene et al., 1960) and Welch t-test (Zimmerman and Zumbo, 1993) are used to test the equality of variances and means respectively. These hypothesis tests are demonstrated for two cases at each height level. The null hypothesis for the Levene

test is "the Standard deviations are equal". The first case, denoted as "Case 1" is to test the $def_{BB}$ scheme against the $NO_{BB}$ (without bright band) one to show the impact of the activation of bright band. The second case, denoted "Case 2" is to test the $rev_{BB}$ scheme against the $def_{BB}$ to check whether the revised scheme degrades or improves the FG departure statistics. If the p-value is less than 0.05, the null hypothesis is rejected, which means that the standard deviations are statistically significantly different. The differences are represented by red and blue triangles. For example (Panel 9b, Case 1), the blue color represents

those height levels where the standard deviation of $def_{BB}$ scheme samples are less than the standard deviation of $NO_{BB}$ (indicate improvement) and reverse for red color (indicate degradation). The gray color represents levels of neutral impact (equal standard deviations). Similarly the Welch t-test is performed, but for the mean.

It is observed that the standard deviations and biases of FG departure statistics with the revised BB ($rev_{BB}$) scheme are improved in the three domains over the ones with the default BB ($def_{BB}$) scheme in the melting zone at Ku band. Even

though to a lesser extent, the impact is also positive at Ka band in the melting layer. From the CFADs that are shown in section 5.3 (see Figures 14 to 17) it is clear that the revised parametrization $rev_{BB}$ provides a more realistic simulation of the bright band in terms of reflectivity profiles compared to both $NO_{BB}$ and $def_{BB}$. The error characteristics in Figure 9 are larger when the $def_{BB}$ scheme is used, followed by $rev_{BB}$ scheme compared to the ones of without any bright-band ($NO_{BB}$) simulations. One possible explanation for the larger standard deviation despite more realistic simulations, compared to the

$NO_{BB}$ scheme, is that activating the melting layer diminishes the overall smoothness of simulated profiles. This can lead to spurious degradations of the FG departure statistics. This can somehow be linked to a double-penalty effect in the vertical, more known on the horizontal when performing precipitation verification (Ebert et al., 2007; Roberts and Lean, 2008). A good example was shown in the subsection 4.2 with the single profile shown in Figure 8. The freezing level height is at approximately 3.5 km in the observations and approximately 4.0 km in the model. Because of the shift of the freezing level

height between observation and NWP model, the FG departure is overestimated just below the freezing level (occurs first either in observation or simulation). Then, the overestimation continues until the bottom of melting layer (occurs lastly either in observation or simulation). This is called a double penalty issue and degrades the forecast scores (Cintineo et al., 2014).





Without any simulation of the bright band ($NO_{BB}$ scheme), there is only one penalty located at the location of the observed BB, which explains the smaller FG departures statistics.

The significance test for Case 1 (e.g. $def_{BB}$ versus $NO_{BB}$) reveals that high magnitudes of standard deviation and bias due to the bright band are statistically significant (shown by red triangles). However in the tropics, there is significant improvement in error over the rainy levels (especially at Ka band). This is due to a too strong bright band which significantly increases the attenuation within the bright band, and in turns reduces the overestimation artificially in the rainy levels. However, the opposite is true for Case 2 (e.g. $rev_{BB}$ versus $def_{BB}$ ) which indicates that errors are better represented by $rev_{BB}$ in comparison to

$def_{BB}$ and are statistically significant. Panels 9d and l show that bias with $rev_{BB}$ are suddenly larger than $def_{BB}$ scheme for the ice levels of the cloud ($\sim$ 5-6 km altitude), and vice versa for lower levels. This is due to the positive (negative) bias in the ice levels (large negative bias). From the cloud top, the bias magnitudes cancel out to zero with $def_{BB}$, but not with $rev_{BB}$ scheme (see Panel 9c), before shifting to large negative bias in and below the melting region.

     One important feature that can also be noted in Figure 9 is the negative bias in the rainy levels, which can be quite large,

especially in the Tropics. Indeed, the bias magnitude is only in 0-5 dB range in the northern and southern hemispheres (top and bottom panels) but can reach approximately -15 dB in the Tropics (middle panels). The bias could arise from several sources, including from the physical parametrization of the model for tropical convection, but also from a misrepresentation of convective rainfall in the forward operator. This study indeed assumes a 5% for convective hydrometeors precipitation fraction. As mentioned in subsection 4.2 and in Appendix B, small precipitation fraction can drastically increase the reflectivity, which

results into a strong overestimation of the reflectivity. This needs to be investigated in a future study. Another assumption is the Particle Size Distributions (PSDs) used for rain. This study uses the Marshall and Palmer PSD for both stratiform and convective rain, but other PSDs could be tested as well. For example, as done by Fielding and Janisková (2020) for the simulation of Cloudsat-CPR reflectivity in the ZmVar forward operator, the Illingworth and Blackman (2002) PSD could be tested for convective rain. For stratiform rain, the Abel and Boutle (2012) PSD, which is the one used in the microphysical

scheme of ARPEGE, could be used to have consistent assumptions in the forward operator and in the model.



**Figure 9.** Standard deviation, and bias of Observed-minus-simulated attenuated reflectivities (or First guess departures) are shown for combined samples of two different months (June, 2020 and January, 2021) at Ku band in (a-d) Northern hemisphere, (e-h) Tropics, and (i-l) Southern hemisphere. Results with $NO_{BB}$, $def_{BB}$, and $rev_{BB}$ scheme are compared. The statistical hypothesis test are represented by the colored triangles to assess the equality of standard deviation (Levene's test) and mean (Welch's t-test). Case 1 illustrates the impact of activating bright band using $def_{BB}$ scheme with reference to $NO_{BB}$, whereas Case 2 shows the assessment of $rev_{BB}$ scheme in comparison to $def_{BB}$ scheme. For case 2, blue (red) triangles indicate that the standard deviation of the $rev_{BB}$ scheme samples are less (larger) than the standard deviation of $def_{BB}$, which indicates an improvement (degradation) of the $rev_{BB}$ scheme. Gray colors represent levels of neutral impact (equal standard deviations). It is to be noted that same behaviour is obtained with separated one month samples.







**Figure 10.** Similar as Figure 9, but for at Ka band





# 5    Application of melting layer for precipitation classification

An application of the combined use of Ku and Ka band attenuated reflectivities is the use of the Dual Frequency Ratio (DFR = ZKu- ZKa) to classify precipitation into three precipitation categories i.e. Stratiform, Convective and Transition (Iguchi et al., 2010; Awaka et al., 2016). If a bright band and rain are detected, the category of a profile will be decided in accordance to

the variation of the DFR profile. In this study, the same DFR classification strategy (Awaka et al., 2016) as the one used in the Level 2 product of GPM DPR observations is applied to ARPEGE simulations. The methodology is described in subsection 5.1; followed by a validation on the same case study as the one used in the subsection 4.2 and on the full period of study as well.

## 5.1    Methodology

Figure 11 shows the flow diagram of the methodology adopted for this study (Awaka et al., 2016). If there is a vertical profile

of DFR from simulated Ku and Ka attenuated reflectivities, the first step is to check if there is a melting region. The melting region is detected if the temperature ($T$) is in between 273 K and 277.5 K, and a DFR larger than 0.0 dB. After detecting the bright band, the maximum of DFR is searched above 1 km and below 2 km from the freezing level height (height at 273 K, denoted as point A) within the melting zone only. If this criteria is satisfied, then the maximum DFR value is identified (hereafter, $DFR_{max}$) and named as point B.

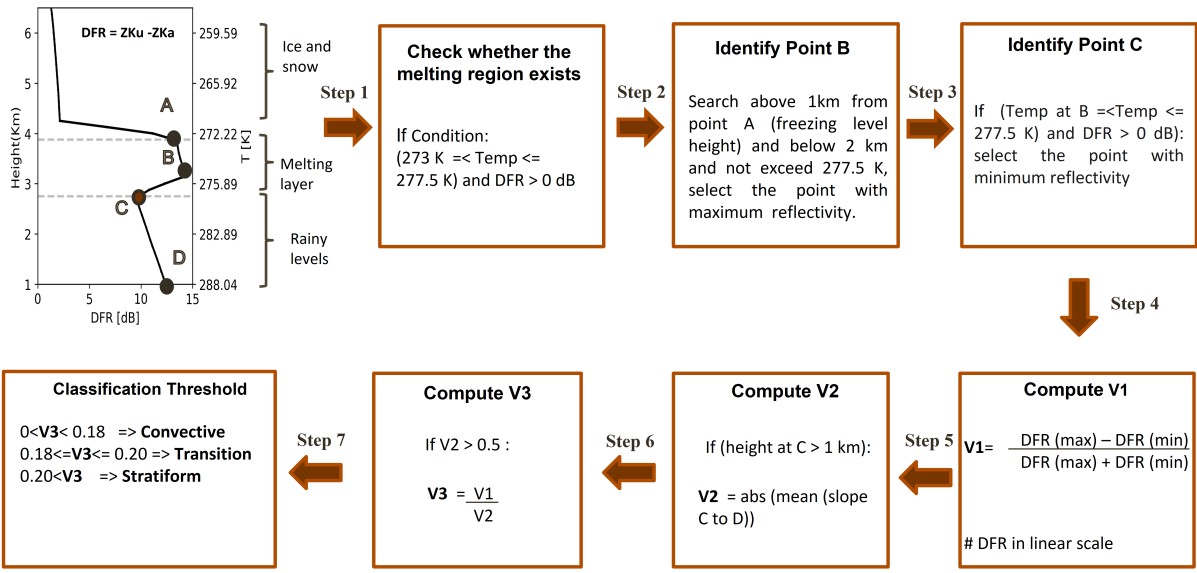

**Figure 11.** Methodology adopted for precipitation type classification using Dual frequency Ratio method, which is based on the detection of the melting layer in the simulations.

The next step is to find the bottom of the melting layer, denoted by point C. This is done by searching for the minimum value of DFR (hereafter, $DFR_{min}$) below the melting layer. The $DFR_{max}$ and $DFR_{min}$ are in dB and converted into linear scale.



A V1 index is defined for classification using $DFR_{max}$ and $DFR_{min}$ values as mentioned in Equation 5. The larger is the V1 value, the larger the chances are to categorize the column as stratiform.

$$V1 = \frac{DFR_{max} - DFR_{min}}{DFR_{max} + DFR_{min}} \tag{5}$$

here, $DFR_{max}$ and $DFR_{min}$ are in linear scale.

The next step is to check whether if rain exists or not. If point C is above an altitude of 1 km, then the second index V2 is computed by extracting all height and DFR values from point C to D (bottom most of the profile). The V2 index (dB/km, see Equation 6) is the absolute value of mean slope of DFR from C to D. Larger is the V2 value, larger is the probability for convective precipitation classification. If V2 is less than 0.5 dB/km, then the DFR classification algorithm is not used as

followed in past literature (Awaka et al., 2016; Iguchi et al., 2010). It is to be noted that both V1 and V2 are normalized values, and independent of height and depth of melting layer (Iguchi et al., 2010).

$$V2 = \left| \frac{\sum_{n=c}^{n=D} DFRslope}{N} \right| \tag{6}$$

with $N$ being the number of levels from C to D.

The last step is to calculate the ratio of V1 and V2 to define an effective parameter V3 (= $\frac{V1}{V2}$) for the classification. The

thresholds used here (given in the methodology diagram) for classification are the same as the ones given in the GPM ATBD and also used in the past studies for the classification of GPM observations (Awaka et al., 2016). These threshold values have been computed from extensive statistical analysis of GPM DPR profiles and airborne radar profiles (Iguchi et al., 2010).





## 5.2 Validation on case study

The DFR algorithm is first applied on the case study (same one as previously shown in subsection 4.2). It has been already
discussed in the previous sections that the $rev_{BB}$ scheme allows to have more realistic bright-band simulations, compared with
the observations. As an extra validation step, this study aims as showing whether if the revised melting layer parametrization can
be used for classifying Stratiform/Convective precipitation with a standard method. In this study, the classification algorithm
is first applied to simulations using both bright-band parametrization schemes, and then compared with the classification of
the observations dataset. It should be noted that only DFR algorithm classified pixels in the observation dataset are compared.
Figure 12 (a-c) shows the vertical cross-section of DFR (observation as well as models), and panels (d-e) show V1 parameter,
(g-h) V2 parameter, and (j-k) V3 parameter for the $def_{BB}$ scheme and $rev_{BB}$ scheme respectively. Panels (l) to (n) show the
classified pixels in the observations, in the simulations with $def_{BB}$ scheme, and with $rev_{BB}$ scheme respectively. For ease of
interpretation, markers *I*, *II*, and *III* are mentioned in panel b, which delineate the surrounding pixels on the left, middle, and
right sides of the panel respectively.

Overall the pattern in Figure 12 b and c is in good agreement with the observations (Figure 12 a). The V1 parameter in
both schemes show that pixels in region *I* have higher values than region *II*. On the contrary, V2 parameter shows opposite
pattern, but in accordance with the algorithm. One can observe that magnitude of V1 and V2 differ with the change in bright-
band scheme. Finally, the deciding parameter (i.e V3) enlarge the differences and lead to categorize region *I* as stratiform
precipitation and region *II* as convective precipitation. However, the majority of region *III* pixels have no classes. The reason
could be associated to high DFR values in the rainy levels. The DFR profile with $def_{BB}$ scheme has not only a too strong
bright band but also have high DFR values in the rainy levels, because of high attenuation at Ka band. Higher is the attenuation
due to the bright band, lower is the rain reflectivity at Ka band and higher are the DFR values in the subsequent levels. One
example of high DFR profile can be seen in the Figure 12 (f and i). The dashed and solid lines correspond to $def_{BB}$ and $rev_{BB}$
scheme respectively. As a result, there is no minima in the DFR (refer the step 3 in Figure 11) at the starting point of rainy
region which ends the algorithm and results in no classification. Therefore, more gaps can be seen with the $def_{BB}$ scheme as
compared to $rev_{BB}$ scheme.

To validate the simulated DFR classification algorithm, which mirrors the observed classification algorithm, the classification
results are compared with model variables. The latitude-height cross-sections of hydrometeors are shown in Figure 13. Both
stratiform and convective contents are smaller in the region *I* and slightly larger in the region *II* and significantly larger in the
region *III*. The large DFR in rainy levels in the Figure 12b correspond to an area with model predicted stratiform and convective
rain together. Overall, the hydrometeor crosssection are in reasonably good agreement with the classification results.

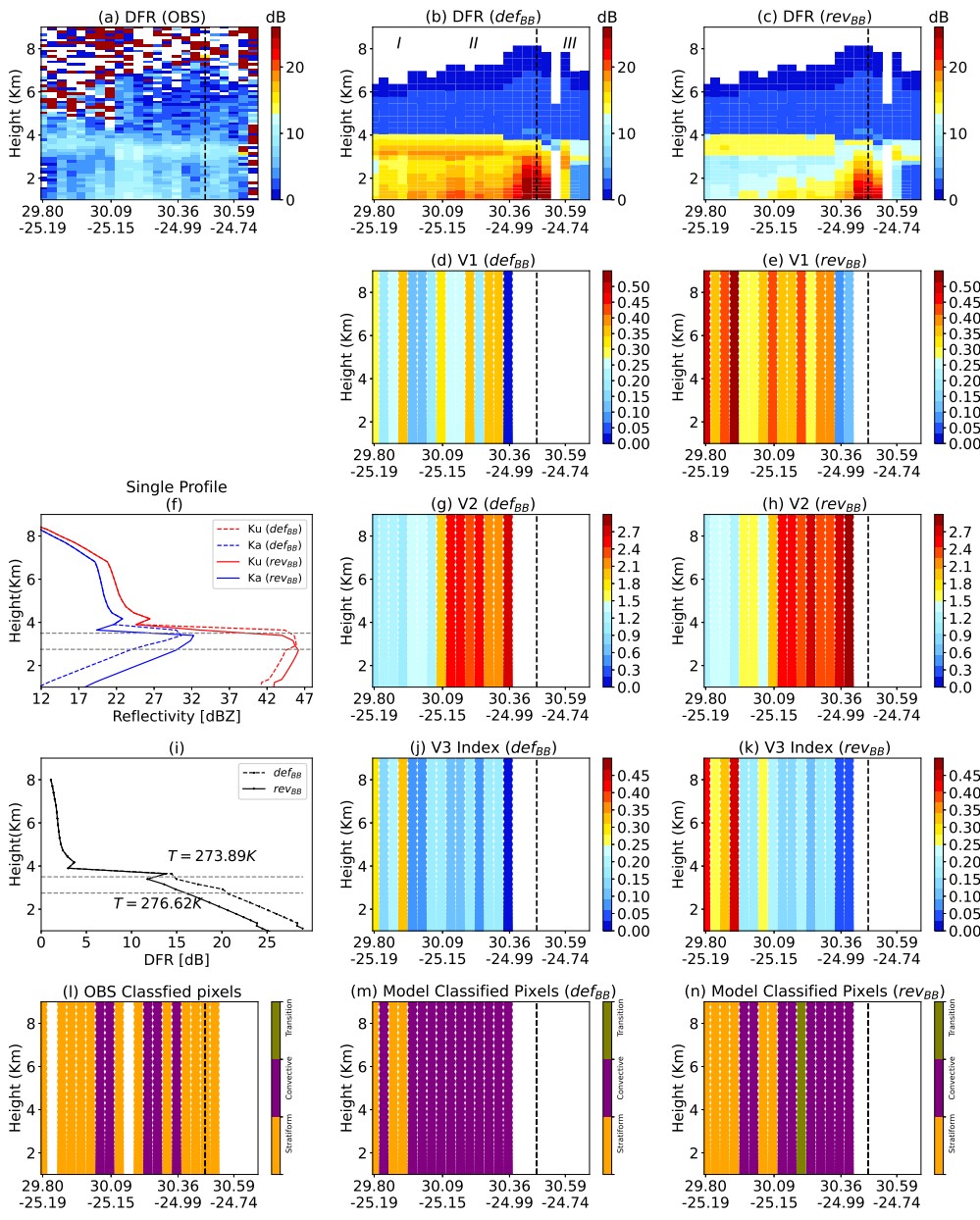

**Figure 12.** (a-c) Vertical cross-section of DFR for the case study in observations and model simulations using $def_{BB}$ scheme and $rev_{BB}$ scheme respectively. (d, g, j) are the V1, V2 and V3 parameters corresponding to the simulations performed using the $def_{BB}$ scheme and similarly, panels (e, h, k) are for $rev_{BB}$ scheme. Panels (l, m, n) are the classified pixels which correspond to the observations, $def_{BB}$ scheme and $rev_{BB}$ scheme respectively. The black dotted horizontal line in panel f and i represents the temperature of top (T= 273.89 K) and bottom (T=276.2 K) of the melting layer for the given profile. Note that we do not perform classification on observation, as this is already performed in the level 2 products. Instead we used classification flag to separate out the DFR classified pixels for comparison. The profile shown in the middle-left side corresponds to the black dashed line in all the panels.



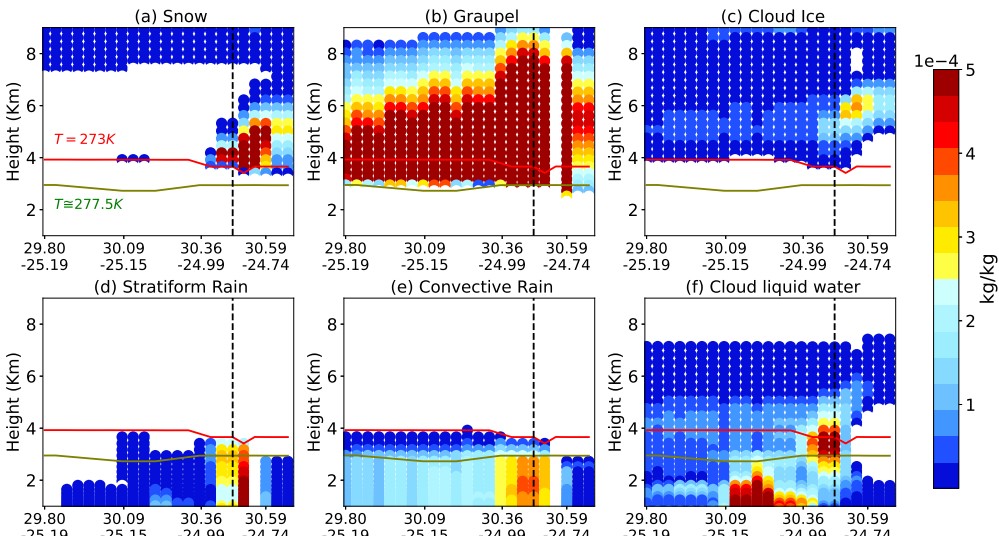

**Figure 13.** Vertical structure of hydrometeors following the cross-section in Figure 12. The red and green lines correspond to the 273 K and 277.5 K temperature levels respectively, which corresponds to the vertical levels in which frozen particles are allowed to melt (see subsection 3.2).





## 5.3 Global analysis over two months period

The classification is now applied to a larger number of samples for two different months (June,2020 and January, 2021) using $def_{BB}$ and $rev_{BB}$ schemes. Figure 14 and 15 shows the normalized Contour Frequency Altitude Diagram (CFAD) of the attenuated reflectivity profiles classified as stratiform in the three geographical domains at Ku and Ka band respectively. The left column is observation, and middle and right columns are the simulations using the $def_{BB}$ and $rev_{BB}$ scheme respectively. The bin size is 2 dBZ x 0.5 km. The vertical distribution of simulated reflectivities are in good agreement with observations at both bands. However, the occurrence of simulated reflectivities are lower over the ice region and significantly higher over the rainy levels in all geographical domains as compared to the observations. It can be seen that the rain reflectivities are 5 dB larger in the simulations than in the observations at Ku band. This reveals the model bias due to microphysical assumptions in ARPEGE global model that leads to overestimate rainfall. It could also be attributed to a mis-representation (e.g. precipitation fraction ) of the large-scale rain in the forward operator. It is worth noting that bright-band peak with $rev_{BB}$ scheme are in better agreement with the observations (especially in the tropics), as compared with the $def_{BB}$ scheme which yields to overestimate of the peak.

Similarly, Figure 16 and 17 show the CFAD diagram for convective precipitation columns. The vertical pattern of convective precipitation is in reasonably good agreement with the observations at Ka band. Even though to a less extent, a good match with the observations is also observed at Ku band. The observed reflectivities at Ku band are in $\sim$ 17-24 dBZ range from the cloud top to the surface over NH and SH regions due to dominance of shallow convection. However, tropical regions are well known for the presence of deep convective systems which lead to strong reflectivities of the order of $\sim$ 35-40 dBZ (Liu and Zipser, 2015; Liu and Liu, 2016). The simulated reflectivities over ice regions are well matched but rainy levels are significantly larger ($\sim$ upto 15-18 dB) over the NH and SH region and $\sim$ upto 5 dB in the tropics. The overestimation of rain reflectivity upto 5 dB is probably associated with the smaller value of convective precipitation fractions in the forward operator (5%) (see Appendix B). One can observe the negative slopes in the lower levels at Ka band (Figure 17), but not at Ku band (Figure 16). This is because of the fact that Ku band radar can easily penetrate the medium to large frozen particles and go deeply to the rainy levels ($\sim$ 1-2 km) whereas, the signal at Ka band is strongly attenuated by these particles.





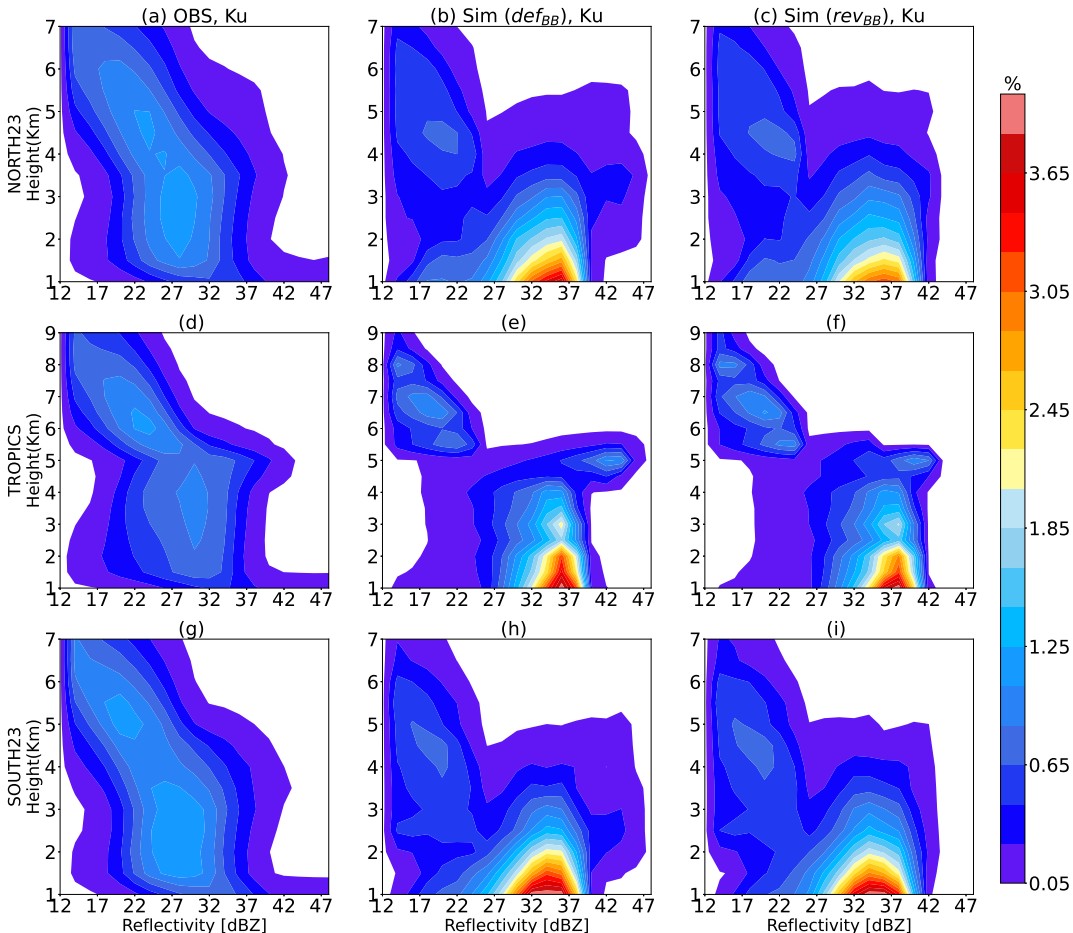

**Figure 14.** Normalized Contour Frequency Altitude Diagram (CFAD) of Ku band for stratiform pixels for the two months period (June, 2020 and January, 2021) over the Northern Hemispheres (top), Tropics (middle) and Southern Hemisphere (bottom). The colorbar represent the percentage of pixels in 2 dBZ x 0.5 km bin.



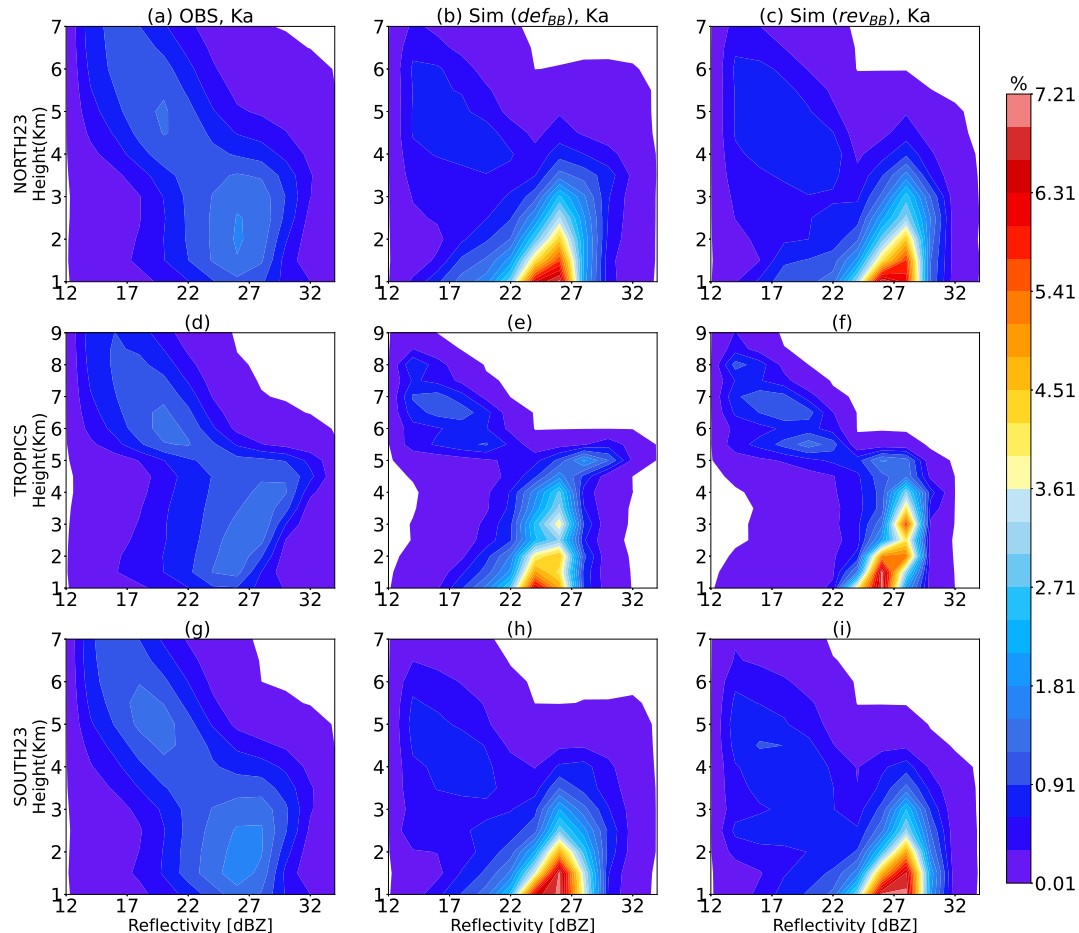

**Figure 15.** Same as Figure 14, but for the Ka band





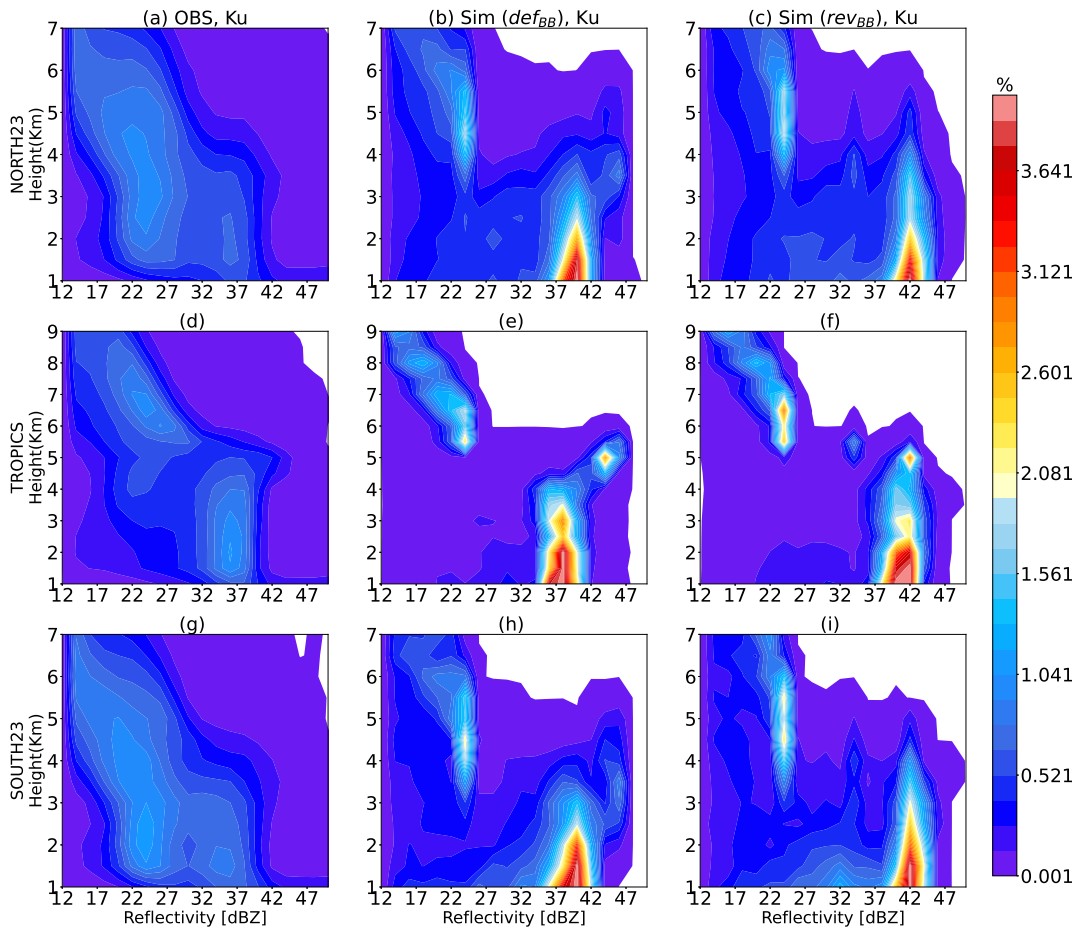

**Figure 16.** Normalized CFAD of Ku band for convective pixels for the two months period (June, 2020 and January, 2021) over the Northern Hemispheres (top), Tropics (middle) and Southern Hemisphere (bottom).





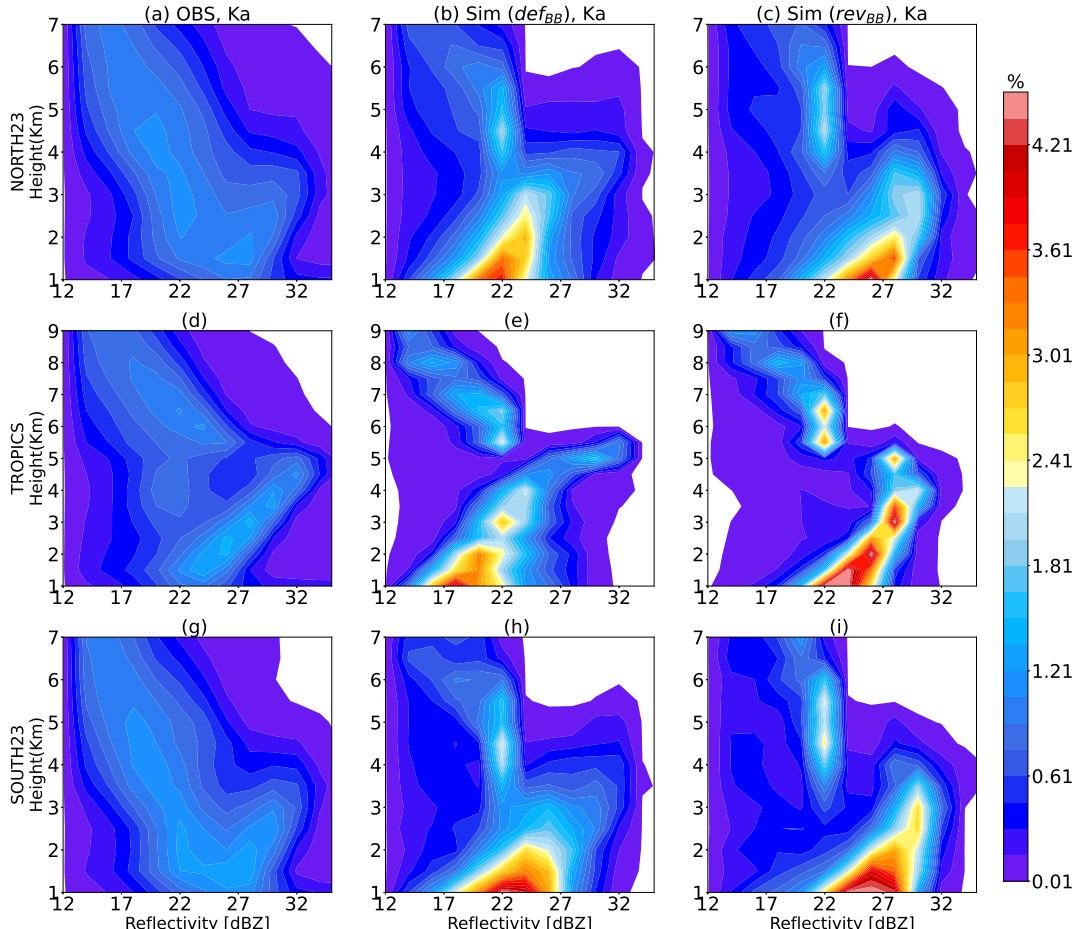

**Figure 17.** Same as Figure 16, but for the Ka band

Table 2 shows the percentage of classified pixels in the observations and ARPEGE NWP model using $def_{BB}$ and $rev_{BB}$ scheme over the NH, tropics and SH regions. In the NH region, the $def_{BB}$ scheme overestimates (resp. underestimates) the number of stratiform (resp. convective) profiles by $\sim$ 13-14 % . However, the $rev_{BB}$ scheme decreases (increases) the number of stratiform (convective) profiles by $\sim$ 3 % in comparison to $def_{BB}$ scheme and tend to move closer to the observations. A similar tendency is observed in the SH. Over the tropical region, both schemes show a good match with the observations for both stratiform and convective precipitations. In summary, the $rev_{BB}$ scheme, as compared to $def_{BB}$ scheme, shows a slight improvement in the detection of the occurrence of stratiform and convective profiles over the NH region. The percentage of transition profiles are very small which do not lead to any robust conclusions.



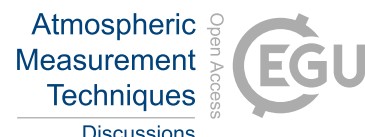

**Table 2.** Percentage of classified pixels in observations and model. The N, T and S symbols denotes northern hemisphere, tropics and southern hemisphere region.

| | | Observations (%) | Model ($def_{BB}$) (%) | Model ($rev_{BB}$) (%) |
|---|---|---|---|---|
| Stratiform | N | 70.03 | 84.90 | 81.74 |
| | T | 75.90 | 76.27 | 76.84 |
| | S | 69.70 | 90.66 | 89.51 |
| Convective | N | 25.37 | 13.66 | 16.69 |
| | T | 20.16 | 21.24 | 20.93 |
| | S | 25.64 | 8.23 | 9.34 |
| Transition | N | 4.59 | 1.42 | 1.55 |
| | T | 3.93 | 2.48 | 2.22 |
| | S | 4.64 | 1.10 | 1.13 |

Table 3 shows the False alarm ratio (FAR) and Probability of detection (POD) values using $def_{BB}$ and $rev_{BB}$ scheme in
the three regions (see details in Appendix C). It can be seen that FAR is low and POD is high for stratiform precipitation and vice-versa for convective precipitation classification. This could be due to the difficulties in precisely locating the convective precipitation in the ARPEGE model, which have a smaller horizontal extent than stratiform precipitation. One can notice that FAR remains the same with both melting schemes for both stratiform and convective events. In case of stratiform event, the $rev_{BB}$ scheme decreases (increases) the POD from 0.84 (0.71) to 0.80 (0.75) over NH and tropical regions, but the impact in
the southern hemisphere is much less important. However, the impact is reversed for convective events. The $rev_{BB}$ scheme improves the classification of stratiform precipitation columns over the tropics, as well as the classification of convective precipitation over the southern hemisphere. Overall, both bright band schemes show great potential in precipitation classification for both qualitative and quantitative assessment.

It should be noted that one of the limitation of the classification algorithm is that multiple scattering effects are not simulated
in the forward operator. In the presence of deep convective core (mostly in Tropics), multiple scattering effects impact the reflectivity simulations and should be accounted for in dual-frequency retrieval algorithm (Battaglia et al., 2015). This study assumes only single scattering effects, which could be one deficiency of the DFR algorithm to detect convective columns in the model space.





**Table 3.** Probability of detection (POD) and False Alarm Ratio (FAR) corresponding to the $def_{BB}$ and $rev_{BB}$ scheme over the three regions respectively.

| | | Stratiform | | Convective | |
|---|---|---|---|---|---|
| | | FAR | POD | FAR | POD |
| | N | 0.3 | 0.84 | 0.66 | 0.18 |
| $def_{BB}$ | T | 0.24 | 0.71 | 0.78 | 0.27 |
| | S | 0.28 | 0.80 | 0.76 | 0.16 |
| | N | 0.31 | 0.80 | 0.71 | 0.18 |
| $rev_{BB}$ | T | 0.24 | 0.75 | 0.77 | 0.24 |
| | S | 0.28 | 0.80 | 0.75 | 0.18 |





# 6 Conclusions and Perspectives

The present study assesses the simulations of GPM/DPR reflectivities within the RTTOV-SCATT model, providing a first validation of the reflectivity scheme that was introduced with RTTOV v13.1. Further, for the first time the optional melting layer parametrization provided with RTTOV is validated in the context of active rather than passive measurements. The ARPEGE global NWP model 6h forecasts are used as input to the forward operator. Simulations are made with and without the melting layer and compared with observations. The RTTOV model offers the Bauer (2001) melting layer parametrization for microwave

radiometers and radars. This parametrization had never been tested for GPM/DPR and in practice does not appear to fit observations well. Hence, the current study proposed a revised version of the Bauer scheme ($rev_{BB}$) to provide a smoother and more accurate vertical representation of melting process in the bright-band.

  The design of the revised bright band stretches the melting layer to 277 K rather than just to 275 K, based on modelling results. The change also introduces an ad-hoc scaling factor that helps reduce backscatter and improves the fit to the observa-

tions. A potential limitation of the scheme is that the results are based on Mie spheres whilst a more physical non-spherical modelling of melting particles is being developed (e.g. Johnson et al., 2016). The next step would be to further enhance this melting layer formulation with one that follows a stronger physical basis, including modelling of non-spherical particles. That may be a challenging task and depends on ongoing developments in the non-spherical modelling, so it is left for future work.

  As a first step of evaluation, results were generated for a case study within the single orbital file on 2nd January, 2021 with

and without a representation of the melting layer. Results indicate that the simulated reflectivities at the freezing level with the $rev_{BB}$ scheme both at Ku and Ka band are closer to the observations (by an order of approximately 5 dB) compared to either the original $def_{BB}$ scheme or no bright band. The simulations were performed for one month periods in two seasons (June, 2020 and January, 2021). The statistical assessment shows significant improvement in the FG departure statistics around the melting layer region using $rev_{BB}$ scheme. This positive improvement is particularly evidenced at Ku band. However, below

the melting layers, the $rev_{BB}$ scheme can lead to an artificial degradation of the statistics which is probably due to a tendency of ARPEGE to produce a too large amount of convective hydrometeors, or to the misrepresentation of convective hydrometeors within RTTOV-SCATT. The $rev_{BB}$ scheme has been made available in the RTTOV v13.2 (released on November, 2022) and the $def_{BB}$ scheme has been discontinued.

  As an indirect validation method, this study applied the methodology for precipitation classification into three categories i.e.

stratiform, convective and transition using the Dual Frequency Ratio (DFR, Ku-Ka) method onto simulations. The performance on case studies is in reasonable agreement with observations and also in accordance with the vertical distribution of hydrometeors produced by the ARPEGE NWP model. Overall, the $rev_{BB}$ scheme shows slightly better potential than $def_{BB}$ scheme in classifying a model column into a given precipitation category. Indeed, the number of classified pixels in each category is in better agreement with the observations (e.g. 25.37 % in the NH for convective precipitation) when the simulations are

performed with the $rev_{BB}$ (e.g. 16.69%) than with the $def_{BB}$ (13.66%) scheme. Finally, CFADs of $rev_{BB}$ classified pixels also reveal the model bias in all hemispheres and show significant overestimation (up to 15 dB) in the rainy regions.





To further improve the quality of the simulations, it is suggested to investigate the free parameters, which lead to the observed overestimation of rain reflectivity. One is precipitation fraction for graupel and convective rain. This study used 5% for all levels (default parameter for passive radiometers operationally used at ECMWF and at Meteo-France). However, subsequent work

suggests a profile of diagnosed convective fraction would have the potential to improve the simulations (not shown here). This should be investigated further in future. Another free parameter is the PSDs for rain. This study used Marshall and Palmer PSD both for stratiform and convective rain, but more sophisticated PSDs (Illingworth and Blackman, 2002; Abel and Boutle, 2012) have been implemented in version 13.2 of RTTOV-SCATT. These PSDs were sucessfully used for the simulation of CPR reflectivities at 94 GHz frequency in the ECMWF IFS model (Fielding and Janisková, 2020, ZmVar) and should therefore be

tested for ARPEGE simulations of GPM/DPR reflectivities.

**Appendix A: Description of the revisited parametrization of the melting layer in RTTOV-SCATT v13.2**

In the original melting layer parametrization ($def_{BB}$), the scattering coefficients $\eta_{T,i}$ at a temperature $T$ are computed according to Equation A1.

$$\eta_{T,i} = \frac{\int_{T_{fl}=273}^{T_{ml}=275} \int_{D_{min}}^{D_{max}} \sigma_i(D,f_m) N_i\{D, WC_i(R)\} dD dz}{z_{layer}} \tag{A1}$$

Here, $i$ is the hydrometeor index of either snow or graupel, $z_{layer}$ (1000 m) is the average width of the melting layer between 273 K and 275 K, dz (10 m) is the discretization step and $f_m$ the fraction of the melted particle of diameter $D$. A melted fraction $f_m$ of 100% (resp. 0%) indicates that the particle is fully melted (resp. frozen). The other parameters used in this equation were introduced in section 3. The melting layer reflectivity $\eta_{T,i}$ is substituted where required in the summation of Equation 1. In this original formulation, the scattering coefficients represent the full melting process between 273 K and 275 K, but they are only

stored in the lookup tables at a temperature $T$ of 273 K. In the revised parametrization $rev_{BB}$, $\eta_{T,i}$ is computed at warmer temperatures from 273 K to 277 K in steps of 1 K, following Equation A2:

$$\eta_{T,i} = \begin{cases} \frac{\int_{T_{fl}}^{T+0.5} \int_{D_{min}}^{D_{max}} \sigma_i(D,f_m) N_i\{D, WC_i(R)\} dD dz}{z_{layer}}, & T = T_{fl} = 273K \\[2ex] \frac{\int_{T-0.5}^{T+0.5} \int_{D_{min}}^{D_{max}} \sigma_i(D,f_m) N_i\{D, WC_i(R)\} dD dz}{z_{layer}}, & 274K \leq T \leq 276K \\[2ex] \frac{\int_{T-0.5}^{T_{ml}} \int_{D_{min}}^{D_{max}} \sigma_i(D,f_m) N_i\{D, WC_i(R)\} dD dz}{z_{layer}}, & T = T_{ml} = 277K \end{cases} \tag{A2}$$

The scattering coefficients at a bin with temperature T are now integrated between T-0.5 K and T+0.5 K temperature ranges. Because of the temperature range of the new melting layer being 273-277 K, the scattering coefficient for the 273 K bin

(when the snow and graupel start melting) are integrated over a range of temperature between 273 and 273.5 K. Similarly, the scattering coefficients for the 277 K bin (bottom of melting layer) are integrated from 276.5 to 277 K. These five temperature bins can be seen as five sub-melting layers in the $rev_{BB}$ parametrization.





The number of temperature levels $N_{levels}$ used to discretized the melting layer is calculated following Equation A3:

$$N_{levels} = \frac{z_{layer}}{dz} \tag{A3}$$

It should be noted that the total number of levels $N_{levels}$ is the same (100 here) in both schemes as the values of $z_{layer}$ and $dz$ are the same. These $N_{levels}$ levels are split across the five temperature bins (ie. five sub-melting layers) of the $rev_{BB}$ scheme. In both schemes, the scattering coefficients are divided by the same average width $z_{layer}$ in equations A1 and A2, regardless of the actual width of the selected temperature bin (1 K in the $rev_{BB}$ scheme and 3 K in the $def_{BB}$ scheme). Therefore, as the actual width of the sub-melting layers of the $rev_{BB}$ scheme are smaller than in the $def_{BB}$ scheme, this division results in an

artificial reduction of the scattering coefficients in the $rev_{BB}$ scheme, as compared with the $def_{BB}$ scheme. This feature was originally introduced unintentionally but because it resulted in good fits to observations it was retained as a result. Were this feature to have been corrected, the results would have been much worse.

  To understand the differences of physical processes between the $def_{BB}$ and the $rev_{BB}$ schemes, the variables of interest

(ie. temperature $T_k$, melted fraction $f_m$ and backscattering coefficient) are inter-compared.

  In both schemes, the temperature profile $T_k$ is calculated according to Equation A4:

$$T_k = T_{fl} + i_{level} * (T_{ml} - T_{fl})/N_{levels} \tag{A4}$$

  Where $i_{level}$ represents a given level within the melting layer depth. The temperature profiles are plotted for the 5 temperature bins (ie. five sub-melting layers) of the revised scheme in Figure A1 for two different diameters (0.35 cm in gray and

and 0.75 cm black plain curves) as a function of the height level $i_{level}$ across the sub-melting layer depth. For each of these levels, the temperature profile is also overlaid in dashed for these two selected diameters for the $def_{BB}$ scheme. These profiles have been plotted at Ku band, and for a hydrometeor content of 1e-4 kg/m3. Figure A1 shows that the temperature increases faster in the $rev_{BB}$ scheme between two successive levels than in the $def_{BB}$ scheme for a given temperature bin. Indeed, as the temperature at the bottom of the melting layer $T_{ml}$ has been increased from 275 K to 277 K in the $rev_{BB}$ scheme, the slope

of the temperature $T_k$ profile is larger in the $rev_{BB}$ scheme than in the $def_{BB}$ scheme (see Equation A4).

  This difference in the temperature profiles impacts directly the fraction of melted particle $f_m$. A larger increase in the temperature values between two successive levels leads to a faster evaporation, and therefore to a faster melting of the particle. This effect is depicted in Figure A2, which shows the vertical profiles of melting fraction $f_m$ across the melting layer depth.

It should be noted that $f_m$ is always initialized at 0 for each of the 5 sub-melting layers in the $rev_{BB}$ scheme, whereas it is continuous across the 100 levels for the $def_{BB}$ scheme. However, despite this difference in their initialization, $f_m$ is not necessarily smaller in the $rev_{BB}$ than in the $def_{BB}$ scheme (see for instance the black curves for the 275 K bin). As it is starting from 0 instead of the value of the previous level in the original formulation, the fraction of melted particle $f_m$ is usually smaller in the revised scheme for the first levels of the selected temperature bin, but then the fraction of melted particle $f_m$ increases





faster in the revised than in the default. For instance, for the 275 K temperature bin in Figure A2, the dashed ($def_{BB}$ scheme) and plain ($rev_{BB}$ scheme) lines cross each other, even though $f_m$ starts from 0 in the $rev_{BB}$ scheme. The same behaviour also appears for the gray diameter at the 274 K. This is due to the fact that the temperature increases faster between two successive levels in the $rev_{BB}$ scheme.

Similarly, the backscattering coefficient profiles are depicted in Figure A3 for both schemes. Figure A3 shows that the backscattering coefficients are smaller at a given level of the $rev_{BB}$ scheme only when their associated melted fraction (shown in Figure A3) at this particular level are also smaller (see for instance the dashed and plain lines for the 275 K bin). As shown in Equation A2, the backscattering coefficients are then integrated over the number of levels of each temperature bin (ie. sub-melting layer). These sub-layer averaged backscattering coefficients (divided by $z_{layer}$) are written in the legend of each

subplot in Figure A3. These integrated coefficients depend on the evolution of the temperature and melted fraction profiles across this particular temperature bin. For instance, the revised scheme yields to larger coefficients for the 274 K and 275 K sub-melting layers as their associated fraction of melted particle is larger at most levels. However, for the 276 K bin, the backscattering coefficient is larger in the default scheme, but only because the differences in the initial values of $f_m$ were too strong so that the revised scheme didn't have time to reach the value of the default scheme. As there is only one bin in the

default scheme (between 273 K and 275 K), the backscattering coefficients of the $def_{BB}$ scheme are averaged over the $N_{levels}$ levels. The corresponding values are 0.00037 (dia = 0.35 cm), and 0.0059 (dia = 0.73 cm). These values are always larger than the sub-melting layer averaged backscattering coefficients of the revised scheme. Consequently, the fact that the $rev_{BB}$ scheme yields to a more realistic bright-band (as compared with the observations) is not only due to the division by the fixed average width $z_{layer}$, but also due to the fact that the integration has been split across 5 sub-melting layers.





**Figure A1.** Profile of temperature ($T_k$) shown for 100 levels inside the melting layer using $def_{BB}$ and $rev_{BB}$ scheme for two different diameters (shown by gray and black curves). These 100 levels are splitted across the five sub-layer temperature bins of the $rev_{BB}$ scheme.







**Figure A2.** Same as Figure A1, but for the melting fractions ($f_m$)







**Figure A3.** Same as above, but for the backscattering coefficients

## Appendix B: Impact of the precipitation fraction on the simulated reflectivities

Figure B1 shows the vertical structure of observations (left panel) and RTTOV simulations (middle and right panels) with a convective hydrometeors fraction $f_i(R)$ of 100 % (middle panel) and of 5% (right panel), for the cloud which is depicted in Figure 7. A convective hydrometeors fraction of 100% (resp. 5%) indicates that the predicted convective hydrometeor covers the full ARPEGE grid (resp. 5% of the grid). As shown by Figure B1, the simulated reflectivity is very sensitive to the convective precipitation fraction value. Figure B1 demonstrates that small hydrometeor fraction $f_i(R)$ significantly increases the reflectivity. The indirect relationship between $f_i(R)$ and particle size distribution (see equations 1 and 3), might N(D) cause this enhancement.





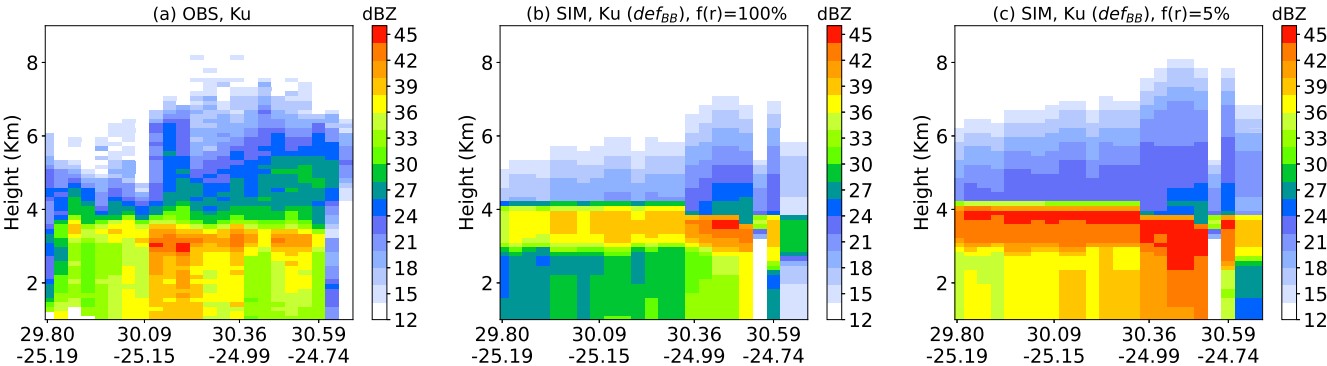

**Figure B1.** Vertical cross section of (a) observation, (b,c) RTTOV simulations with convective precipitation fraction profiles of 100 % (middle) and 5 % (right) for the same cloud as the one shown in Figure 7.

**Appendix C: Contingency Table**

A contingency table is constructed as shown in Table C1. It is a matrix that measures the accuracy between observed and model classified profiles. The diagonal elements **A**, **E** and **I** are the number of profiles which belong to same class in both observation and model. The non-diagonal elements are the mis-classified profiles. For example, element **B** represents the number of profiles, classified as *'stratiform'* in observations, but *'convective'* by the simulations and similarly for others. The following categorical statistics are used:

1. Probability of detection (POD) is a measure of success of algorithm that correctly classify the profile into the *'stratiform'* or *'convective'* category.

2. False alarm ratio (FAR) determines the fraction of mismatched classified profiles between observation and model for a given classification category.

High POD and low FAR indicates higher accuracy.Equation C1 and Equation C2 below illustrate the POD and FAR for stratiform events. Similarly, they are also computed for convective events. It is noted that transition profiles are used in the computation for POD and FAR but the vertical structure is not discussed here because of the low occurrences mentioned in Table 2.

**Table C1.** Contingency table for the occurrence of stratiform, convective and transition pixels

| | | Observations | | |
|---|---|---|---|---|
| | | Stratiform | Convective | Transition |
| Model | Stratiform | A | D | G |
| | Convective | B | E | H |
| | Transition | C | F | I |



$$POD_{stratiform} = \frac{A}{(A+B+C)} \tag{C1}$$

$$FAR_{stratiform} = \frac{D+G}{(A+D+G)} \tag{C2}$$

*Code availability.* The RTTOV-SCATT simulator is available on the EUMETSAT SAF website:https://nwp-saf.eumetsat.int/site/software/rttov/rttov-
620  v13/

*Data availability.* Data can be made available on request.

*Author contributions.* This work was carried out by RM as part of his post-doctoral position under the supervision of MB and PC. AG
provided a scientific and technical support through the entire study. Based on the results of this study, AG and JH also implemented the code
changes in version 13.2 of RTTOV-SCATT. JH provided great support on the use of RTTOV-SCATT. All co-authors collaborated, interpreted
625  the results and wrote the paper.

*Competing interests.* The authors declare that they have no conflict of interest.

*Acknowledgements.* EUMETSAT Satellite Application Facility on Numerical Weather Prediction (NWPSAF) is acknowledged for funding
this work. Our sincere thanks to Emma Turner for providing the DPR coefficients within the RTTOV software. The RTTOV v13.2 is freely
available on https://nwp-saf.eumetsat.int/site/software/rttov/download/. Thanks to Goddard Distributed Active Archive Center (GES DISC
630  DAAC) for providing the GPM/DPR observations and are freely available at https://mirador.gsfc.nasa.gov/. Niels Bormann, Steve English,
Tony McNally, Stefano Migliorini and Mark Fielding are acknowledged for their internal reviews which significantly help to improve the
paper.





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
