# Peer review of "Assessment and application of melting layer simulations for spaceborne radars within the RTTOV-SCATT v13.1 model"

_Atmospheric Measurement Techniques, 2024_

## Referee Comment (RC2)

General Comments:

I appreciate the significant contribution this work makes to radar remote sensing and its potential to enhance the assimilation of radar observations in NWP models. While the radar equations appear to be sound, I have some concerns regarding the use of NWP profiles for evaluation in this context. There are noticeable discrepancies between the simulations and observations, yet the authors seem to overlook these differences, suggesting that there is good agreement, despite the figures indicating otherwise. Additionally, the text would benefit from revisions for clarity and consistency, as some sections can be challenging to follow.

Page 2 L38: Some recent CRTM references would be more suitable here, including:

https://doi.org/10.1175/BAMS-D-22-0015.1

https://doi.org/10.1109/TGRS.2023.3330067

Page 3 L70: you can still see bright band in CloudSat CPR frequencies, e.g., see
https://doi.org/10.1109/TGRS.2023.3330067

P5 L130: 30 minutes seems too high for cloud and precipitation related collocations. Any comment on how this would impact the results?

P6 L138: do you also interpolate when there is gap in reflectivities? That can be quite problematic.

P6 L146: "set of specific number of hydrometeors": do you mean ensemble of hydrometeors?

P6 Equation: I am confused by fraction "f" – isn't that already being represented in the water content values? Is this the maximum overlap probability? We are looking at individual layers when calculating the reflectivity so why would we need to worry about maximum overlap?

On Page 9, Line 225, and in the last paragraph on Page 11, the text is difficult to understand. Overall, the manuscript would benefit from editing for clarity and language.

P14 L279: Please rewrite this paragraph for clarity.

P14 L280-300: it states around L280 that "overall the spatial structure of simulated cloud is well …". I don't really think so. It clearly shows that the structure of clouds in simulations and observations are quite different. Again, around L288, it states the same for vertical structure of clouds but again I disagree with the statement as there is clearly a large

discrepancy between vertical structure of simulations and observations. This is likely due to error in input profiles.

P14 L292: "(e.g., spherical …)" I think the role of input profiles is extremely important here and should be emphasized

P14 L297: this should be better discussed and how it impacts the results. The NWP profiles may not be even suitable for this kind of evaluation studies.

---

## Author Comment (AC1)

**Responses to reviewer 2:**

**General Comments: I appreciate the significant contribution this work makes to radar remote sensing and its potential to enhance the assimilation of radar observations in NWP models. While the radar equations appear to be sound, I have some concerns regarding the use of NWP profiles for evaluation in this context. There are noticeable discrepancies between the simulations and observations, yet the authors seem to overlook these differences, suggesting that there is good agreement, despite the figures indicating otherwise. Additionally, the text would benefit from revisions for clarity and consistency, as some sections can be challenging to follow.**

**Response:**

The authors are grateful to Reviewer 2 for his/her in-depth review of the manuscript and his/her constructive comments which significantly improved the manuscript. We appreciate the concern regarding the input profiles taken from the global NWP model ARPEGE, which has been operational at Meteo-France since 1992. The authors recognize that this concern is particularly true in the context of a global NWP model in which the convection is parametrized and for which the effective resolution of the model is larger than the observations. Therefore, the differences between the model and the observations can indeed arise from several sources such as the modeling of the radiative transfer within the forward operator (e.g. Particle Size Distribution, precipitation fraction, shapes, etc...), as well as modelling of the clouds and precipitation within the forecast used as input. One alternative to reduce the latter source of error in the comparison would be to use cloud and precipitation retrievals as inputs of the radiative transfer (Johnson et al. 2016). This is kept in mind for future work and the authors added some sentences in the manuscript to highlight this particular point. Nonetheless, the authors think that the present work and analysis between observations and simulations is important as it is a preliminary step before the assimilation of these observations in a global NWP model, which has proven to be useful by several NWP centers (Ikuta et al. 2021; Fielding et al. 2020).

Please, you can find in the next pages our point-to-point responses, along with the revised version of the article (changes have been made in red in the text). Besides, the sections which were difficult to follow have been modified (changes have been made in blue in the text).

**Comment 1:**

**Page 2 L38: Some recent CRTM references would be more suitable here, including: https://doi.org/10.1175/BAMS-D-22-0015.1**

https://doi.org/10.1109/TGRS.2023.3330067

**Response 1:** The authors thank the reviewer for sharing the references. The authors included the above references in the revised manuscript in the introduction.

**Comment 2:**

**Page 3 L70: you can still see bright band in CloudSat CPR frequencies, e.g., see https://doi.org/10.1109/TGRS.2023.3330067**

**Response 2:** The authors thank the reviewer for this comment. The paragraph has been slightly modified.

**Comment 3:**

**P5 L130: 30 minutes seems too high for cloud and precipitation related collocations. Any comment on how this would impact the results?**

**Response 3:** In this study, the authors wanted to remain close to the temporal window of the 4DVar data assimilation system of ARPEGE, in which observations are assimilated within a 6-h assimilation window divided in 11 time slots of 30 minutes (+-15 minutes), and 2 time slots of 15 minutes. For each time slot, all the observations are assimilated as if they were valid at the time of the centre of the time slot. To increase the number of samples, in our study we decided to slightly increase this temporal collocation window from +-15minutes to +-30 minutes. The authors think that this temporal window is a good compromise to gather a sufficiently large number of cloudy and precipitating observations which are still valid at the forecast time. The authors added this sentence in the dedicated section of the manuscript.

Please note that to take into account the spatial and temporal mismatches between observations and simulations, the statistics have only been calculated when a cloud or precipitation system is observed and simulated.

**Comment 4:**

**P6 L138: do you also interpolate when there is gap in reflectivities? That can be quite problematic.**

**Response 4:** The authors thank the reviewer for this question. During the interpolation step, when a gap in the reflectivities is found, it is indeed preserved in the interpolated data. The FG departure is computed only when both observed and simulated reflectivities have value above radar sensitivity at the same height bin, otherwise not considered for statistical analysis. To make it

clearer for the reader, one sentence was added in the reviewed manuscript after the sentence highlighted by the reviewer.

**Comment 5:**

**P6 L146: "set of specific number of hydrometeors": do you mean ensemble of hydrometeors?**

**Response 5:** Exactly, the authors meant "an ensemble of hydrometeors". RTTOV 13.1 can allow any arbitrary set of hydrometeors and this study considers an ensemble containing the 6 hydrometeors predicted by ARPEGE (rain, convective rain, snow, graupel, cloud water and cloud ice). The total radar reflectivity is then the summation of the reflectivity computed from each of the six hydrometeors (as mentioned in equation 1). The authors have changed "set of specific number of hydrometeors" to "an ensemble of hydrometeors" in the text.

**Comment 6:**

**P6 Equation: I am confused by fraction "f" – isn't that already being represented in the water content values? Is this the maximum overlap probability? We are looking at individual layers when calculating the reflectivity so why would we need to worry about maximum overlap?**

**Response 6:** In RTTOV-SCATT, the hydrometeor contents are normalized by a hydrofraction based on Geer et al. (2009) for passive microwave instruments. When radar capability was added to RTTOV-SCATT, a specific normalization was added. In RTTOV-SCATT V13, a hydrometeor fraction profile needs to be specified as input for each hydrometeor type.

Then, the hydrofraction ($f(R)$) variable serves two purposes in the radar equation. (1) to normalise the hydrometeor content ($W(R) = W_{av,i}(R) / f_i(R)$, see equations 2 and 4) by the fraction occupied by each hydrometeor over the grid cell. This is particularly important for global NWP models (such as ARPEGE) for which the resolution ranges from 5km to 25km. The normalised content is then used to derive reflectivity and extinction coefficients from the look-up table. (2) The hydrometeor fraction $f(R)$ is then used to denormalize the scattering/extinction coefficients. Please note that this coefficient is only used for "hydrometeor-related" bulk scattering coefficients. The attenuation implied by the presence of gas are not affected by the fraction (see eq.3)

**Comment 7: On Page 9, Line 225, and in the last paragraph on Page 11, the text is difficult to understand. Overall, the manuscript would benefit from editing for clarity and language.**

**Response 7:** The text has been revised. For the English, the authors would like to point out that the manuscript was originally reviewed by two native English speaking co-authors. It has also been internally reviewed by 5 native English speaking colleagues from ECMWF, Météo-France and UK Met-Office. Nonetheless, the authors agree that some sections were difficult to follow. As an additional proofreading, the English has been reviewed by an internal native speaking colleague from Météo-France. The English track changes can be seen in blue in the revised manuscript.

**Comment 8:**

**P14 L279: Please rewrite this paragraph for clarity.**

**Response 8:** As the reviewer suggested, the authors revised the paragraph.

**Comment 9:**

**P14 L280-300: it states around L280 that "overall the spatial structure of simulated cloud is well …". I don't really think so. It clearly shows that the structure of clouds in simulations and observations are quite different. Again, around L288, it states the same for vertical structure of clouds but again I disagree with the statement as there is clearly a large discrepancy between vertical structure of simulations and observations. This is likely due to error in input profiles.**

**Response 9:** The authors agree with the reviewer's comment that there are discrepancies between the observed cloud and the forecasted cloud structures. This is expected as we are considering a global model with parameterized convection. We have carefully checked and there is no error in the input profiles. To address the reviewer's comment, we have erased the sentences which were stating that the simulation was well representing the observed cloud structure.

**Comment 10:**

**P14 L292: "(e.g., spherical …)" I think the role of input profiles is extremely important here and should be emphasized.**

**Response 10:** The authors agree that it should be clearer in the text that a source of errors is also the forecast model. The sentence highlighted by the reviewer was modified to emphasize this.

**Comment 11: P14 L297: this should be better discussed and how it impacts the results. The NWP profiles may not be even suitable for this kind of evaluation studies.**

**Response 11:** The authors agree with the reviewer's comment and we have added a dedicated paragraph at the end of the conclusion regarding this limitation of the study. However, the authors believe that the NWP profiles are important for this kind of evaluation study because the analysis of differences between observations and simulations are the first step prior to assimilation.

*Fielding MD, Janisková M. Direct 4D-Var assimilation of space-borne cloud radar reflectivity and lidar backscatter. Part I: Observation operator and implementation. Q J R Meteorol Soc. 2020; 146: 3877–3899. https://doi.org/10.1002/qj.3878*

*Geer, A. J., P. Bauer, and C. W. O'Dell, 2009: A Revised Cloud Overlap Scheme for Fast Microwave Radiative Transfer in Rain and Cloud. J. Appl. Meteor. Climatol., **48**, 2257–2270, https://doi.org/10.1175/2009JAMC2170.1.*

*Ikuta Y, Okamoto K, Kubota T. One-dimensional maximum-likelihood estimation for spaceborne precipitation radar data assimilation. Q J R Meteorol Soc. 2021; 147: 858–875.*
*https://doi.org/10.1002/qj.3950*

---

## Author Comment (AC2)

**Responses to Reviewer 1**

**General Comments:**

**The radar simulator in RTTOV is a very important extension and makes the model even more usable for the community. Having a bright-band scheme included in the model is very good as well. Therefore, the manuscript gives a substantial contribution to the scientific progress. Technically, I would have appreciated an additional proofreading, maybe even by a native speaker, before submitting the manuscript to AMT for discussion. Some parts or sentences are not very easy to read. In addition, there are many inconsistencies:**

- **the use of commas before and in enumerations**
- **whether bright-band is written "bright-band" or "bright band"**
- **same for Ku- and Ka-band**
- **using in units in situations like "15 dBZ and 10 dBZ" or "15 and 10 dBZ"**
- **acronyms they are sometimes introduced like "Global Precipitation Mission (GPM)" and sometimes "global precipitation mission (GPM)"**

**In the specific comments, I'm not mentioning all typos and, in my opinion, falsely set commas.**

**General response:**

The authors would like to thank the reviewer for its careful review of the manuscript and for providing valuable suggestions which significantly improve the quality of the work. In accordance with the suggestions, the authors have thoroughly checked the manuscript and shared the point by point responses to the comments by Reviewer 1 herewith. In the revised pdf file, the changes are highlighted in orange. For the English, the authors would like to note that the manuscript was originally reviewed by two native english co-authors. It has also been internally (ECMWF, Météo-France, UK Met-Office) reviewed by 5 native english speaking colleagues. Nonetheless, the authors agree  that some sections were difficult to follow. Therefore, as an additional proofreading, the English has been reviewed by the one native speaking colleague from Météo-France. The English track changes can be seen in blue in the revised manuscript.

**Specific comments:**

**Comment 1: p.1 l.4: TOVS is not explained here**

**Response 1:** The authors expand the TOVS as TIROS Operational Vertical Sounder in line 4 in the revised manuscript.

**Comment 2: p.1 l.9: GPM  is not explained here**

**Response 2:** In the revised manuscript, the full acronym of GPM as Global Precipitation Measurement is written line 9.

**Comment 3: p.2 l.24: CloudSat should be written with capital S throughout the manuscript**

**Response 3:** The authors thank the reviewer for this suggestion. The authors have rewritten the CloudSat with capital S everywhere in the revised manuscript.

**Comment 4: p.2 l.45-47: The sentence should be rewritten to "In the melting layer, the maximum size snowflakes are first transformed into wet flakes and then to raindrops of smaller sizes of equivalent mass and less number density as compared to the original flakes (Galligani et al., 2013).". Thereby, I'm not sure about "less number density". Why should that change? You do not change the number of particles within the volume.**

**Response 4:** The authors thank the reviewer for this comment and the sentence has been changed. According to Galligani et al., (2013)  "the flakes collapse into raindrops of much smaller size than the original flakes of the same mass, but with an increased fall speed, yielding smaller particle concentration.". A similar statement can also be found in Rupayan Saha and Firat Y. Testik, (2023).

*Galligani, V. S., C. Prigent, E. Defer, C. Jimenez, and P. Eriksson (2013), The impact of the melting layer on the passive microwave cloud scattering signal observed from satellites: A study using TRMM microwave passive and active measurements, J. Geophys. Res. Atmos., 118, 5667–5678, doi:10.1002/jgrd.50431.*
*Saha, R., & Testik, F. Y. (2023). Assessment of OTT Parsivel 2 Raindrop Fall Speed Measurements. Journal of Atmospheric and Oceanic Technology, 40(5), 557-573.*

**Comment 5: p.3 l.63: change "retains" to "retain". Geer and Bardo are two people.**

**Response 5:** The authors thank the reviewer for the grammatical correction which has been changed.

**Comment 6: p.3 l.79: Why are graupel and hail written capitals? Same later on in the manuscript.**
**Response 6:** The authors change to small letters in the revised manuscript.

**Comment 7: p.3 l.87: "whether if" remove "if" p.6 l.139: Remove "." in "mm6.m-3"**
**Response 7:** The authors thank the reviewer for the corrections. We removed the 'if' and '.' in the revised manuscript.

**Comment 8: p.6 l.149: "radar range gate"**
**Response 8:** The authors change 'radar gate' to 'radar range gate' in the revised manuscript.

**Comment 9:  p.7 l.181: "The Marshall and Palmer (1948)(hereafter MP)" what?**
**Response 9:** The authors forgot to mention the 'PSD' word here. It has been incorporated in the revised manuscript.

**Comment 10: p.8 Table1: Remove "s" from hydrometeors**
**Response 10:** The 's' is removed from Table 1 in the revised manuscript.

**Comment 11: p.9 l.223: "density and mass ... are"**
**Response 11:** The authors thank the reviewer for spotting out the grammatical mistake, which has been corrected in the revised version.

**Comment 12: p.9 l.226: "...melted hydrometeor fraction f_m..."**
**Response 12:**  The authors made the correction in the revised manuscript.

**Comment 13: p.9 l.231: "...has also been used..."**
**Response 13:** The authors thank the reviewer for highlighting the grammatical mistake, which has been corrected in the revised paper.

**Comment 14: p.10 l.243: "...melting layer model" or "scheme"**
**Response 14:** To avoid the confusion, the authors have written 'melting layer scheme and two-phase model' in the revised manuscript.

**Comment 15: p.12/13 figure 3/4: Why do the lines at different temperatures cross without following any rules.**
**Response 15:** The authors would like to thank the reviewer for this interesting question. To answer this particular point, the authors investigated why the reflectivity is larger (resp. smaller) at 273K than at 277K for a content of $2*10^{-5}$ kg/m3 (resp. 10-3 kg/m3). This non-linear behaviour is due to the evolution of the melting process, which is different at 273K than at 277K.

Indeed, in the next two figures, the authors show the variables which play a key role in the computation of the reflectivity inside the melting layer, for the two different temperature bins. The

variables are shown as a function of the level inside each sub-melting layer on the y axis, and as a function of the diameter (x axis). The colorbar represents one variable of interest. The results are shown at Ka band only. They are shown on the left columns for the 273K bin, and on the right columns for the 277K bin. The authors have plotted:

- **The diameter of the melted particle (top panels a and e):** This diameter can be seen as the remaining frozen part of a given melted particle. It is equal to the frozen diameter at the top of the sub-melting layer, and it is equal to 0 (red colours) when the particle is fully melted. A given vertical column of the two-dimensional plot represents the melting process for a particular particle of diameter D across the sub-melting layer. For the T273K bin (top left panel, a), the melting process is slower than for the 277K bin (top right panel, e) because of the colder temperature. Therefore, the melted diameters are always almost equal to the frozen diameter for the 273K bin, except for the smallest diameters at the end of the 273K sub-melting layer. However, the melting process is quicker at 277K (see e panel) as more particles have their melted diameters which reach values close to 0 (red values).
- **The number density of the melted particles in panels b and f**.
- T**he reflectivity** *before* **integrating over the PSD and the height levels** (it is displayed in dBZ in panels c and g):  This value corresponds to the backscattering cross-section, multiplied by the melted number density (for the melted diameter d_melt), and by the square value of the melted diameter (the equation is given in the subtitle). Therefore, when a particle is melted (i.e. its melted diameter equals 0), then its associated backscattering cross-section (for a given diameter and a given level) is close to 0 because the backscattering cross-section is multiplied by the diameter of the remaining frozen part of the particle. Therefore, wherever (in terms of levels and diameters) a particle is melted, its contribution to the overall reflectivity is very small (see for instance the area displayed by the black circle).
- **The reflectivity** *after* **integrating over the PSD and the height levels (panels d and h):** in the 2D plot, one can see the reflectivity that is being accumulated over the PSD and the height levels across the 273K bin (d) and the 277 K bin (h). The value on the bottom right corresponds to the value which is stored in the look-up table for this specific bin. For a content of 2*10-5 kg/m3, the final value (about 0~dBZ) is larger for the 273K bin, compared to the value of the 277K bin (about -3dB). This is because the areas in which the particles are completely melted at 277K, corresponds to levels in which they are not melted at all at 273K. Therefore, their contributions to the overall reflectivity is larger at 273K than at 277K, especially as there is a very large number of small particles for this content. This behaviour is not observed for larger content because of the PSD (see for instance the comparisons of the number density plots between a content of 10-3 kg/m3 and a content of 2*10-5 kg/m3). Indeed, the number of larger (resp. smaller) particles is larger (resp. smaller) for larger contents, and these large particles are not fully melted in most of the levels of the different

sub-melting layers. Therefore, we end up with a more logical behaviour, with larger reflectivity at 277K than at 273K.

The authors added a sentence to briefly describe this behaviour in section 4.1. The authors recognise that the assumptions made on the evolution of the melting process within each sub-melting layer has a strong impact on the subsequent simulated reflectivities. To avoid this issue, an alternative could be to add the melted fraction (calculated based on the liquid and frozen contents of the NWP model) as a predictor in the lookup tables (in addition to the temperature and hydrometeor content), as it is done in the T-matrix ground-based (Augros et al. 2016) and airborne cloud (Borderies et al. 2018) radar forward operator. This requires a lot of work and is therefore left for future work.

*Variables of interest for a content of 2*10-5 kg/m3 at Ka band:*

[Figure]

[Figure]

**Comment 16: p.20 l.345pp: "this can.." and "This can..."**
**Response 16:** The authors revised the line.

**Comment 17: p.21 l.361/362: I do not understand this sentence.**
**Response 17:** The authors agree that the sentence was hard to understand and didn't provide any new relevant information. The sentence has been removed.

**Comment 18: p.22 figure 9: "...less (larger)..." less and larger do not belong together. It is either "less and more" or "smaller and larger"**
**Response 18:** The authors thank the reviewer for the correction. We corrected "smaller and larger" in the revised manuscript.

**Comment 19: p.24 l.384: There is always a profile of DFR. Sometimes the values are just equal 0.**
**Response 19:** Thank you for pointing out. To avoid the confusion, the sentence has been revised.

**Comment 20: p.25: First define V1 and V2 before talking about them.**

**Response 20:** The authors agree with the reviewer and have modified the text

**Comment 21: p.25 l.405: "(given in the methodology diagram)" Please mention the threshold values here.**

**Response 21:** As suggested, we added the threshold values.

**Comment 22: p.26 l.418: Mention here what the markers I, II, and III denote.**

**Response 22:** To avoid the confusion, the sentence has been revised with a clearer explanation of marker I, II and III.

**Comment 23: p.26 l.426/427: This sentence needs to be revised by including some articles.**

**Response 23:** As suggested, the sentence has been revised by adding a reference.

**Comment 24: p.27 figure 12: labels on the colorbar need to be adjusted better.**

Response 24: We revised the figure with clear visibility of the colorbar.

**Comment 25: p.29 l.449: I do not see the bright-band peak in the observations.**

**Response 25:** The authors agree that the bright-band peak is not visible in the CFADs in the mid-latitudes, but only in the Tropics. Indeed, as shown in Figure 14 (d) and 15 (d), the bright band peak is visible around 5 km in the observations in the tropics. The authors have revised the sentence in the manuscript.

**Comment 26: p.29: In line 133 you mention the averaging of the profiles and promise to discuss it later. I would have expected this here.**

**Response 26:** The authors thank the reviewer for pointing out this. As the results are similar (in CFADs as well as on the categorical scores), we didn't discuss it. According to your suggestion, the authors added one sentence in section 5.3 (from line 464). *"These discrepancies are similar in the three geographical domains, indicating that the differences of horizontal resolution across the globe with the ARPEGE stretch grid has a secondary effect with respect to other model biases."*

**Comment 27: p.36 l.487: Please give a reference where the melting layer scheme is validated for the passive simulations.**

**Response 27:** The authors thank the reviewer for this suggestion. As suggested, a reference (Bauer, 2001) has been added.

**Comment 28: p.38 l.544: Why are there only five temperature bins but n_levels level?**

**Response28:** This study is based on the original formulation of Bauer (2001), which was originally introduced in RTTOV. One important feature of this model is that the fraction of melted particles is not specified as input parameter. Therefore, the melting layer is subdivided into a given set of height profiles (nlevels). This subdivision is intended to allow a smooth vertical representation of the melting process, starting from the completely frozen particle, until fully melted. The resulting bulk scattering coefficients are then integrated over these n_levels heights. In the original formulation, only the 273 temperature bin was specified, even though the scattering coefficients were integrated between 273K and 275K, with a melting process which was discretized within n_levels. In this revised parametrization, it has been decided to slightly change the parametrization to get closer to the NWP models (in which we can have outputs at all temperatures), by adding 5 temperature bins in total, and by keeping the original subdivision of n_levels across the full melting layer. The authors added one sentence into the appendix to mention this point L573.